

# Inter-comparison of the Elemental and Organic Carbon Mass Measurements from Three North American National Long-term Monitoring Networks

Tak W. Chan[1,*], Lin Huang[1,*], Kulbir Banwait[2], Wendy Zhang[1], Darrell Ernst[1], Xiaoliang Wang[3], John G. Watson[3], Judith C. Chow[3], Mark Green[3], Claudia I. Czimczik[4], Guaciara M. Santos[4], Sangeeta Sharma[1], Keith Jones[5]

[1] Climate Chemistry Measurements and Research, Climate Research Division, Environment and Climate Change Canada, 4905 Dufferin Street, Toronto, Ontario, Canada, M3H 5T4

[2] Measurements and Analysis Research Section, Air Quality Research Division, Environment and Climate Change Canada, 4905 Dufferin Street, Toronto, Ontario, Canada, M3H 5T4

[3] Division of Atmospheric Sciences, Environmental Analysis Facility, Desert Research Institute, 2215 Raggio Parkway, Reno, NV 89512

[4] Earth System Science, University of California, Irvine, CA 92697-3100, USA

[5] Applied Environmental Prediction Science Pacific & Yukon, Prediction Services Operations West, Prediction Services Directorate, Meteorological Service of Canada, #201-401 Burrard Street, Vancouver, B.C., Canada, V6C 3S5

* Corresponding authors, Email: tak.chan@canada.ca, Phone: (416) 739-4419; lin.huang@canada.ca, Phone: (416) 739-5821

**Keywords**

Black carbon, thermal evolution, air pollution, carbonaceous aerosol, IMPROVE, CAPMoN, CABM



## Abstract

Carbonaceous aerosol is a major contributor to the total aerosol load and being monitored by diverse measurement approaches. Here, ten years of continuous carbonaceous aerosol measurements collected at the Centre of Atmospheric Research Experiments (CARE) in Egbert, Ontario, Canada on quartz filters by three independent networks (Interagency Monitoring of PROtected Visual Environments (IMPROVE), Canadian Air and Precipitation Monitoring Network (CAPMoN), and Canadian Aerosol Baseline Measurement (CABM)) were compared. Specifically, the study evaluated how differences in sample collection and analysis affected the yield of total carbon (TC), organic carbon (OC), and elemental carbon (EC). When all measurements were normalized with respect to concentration measured in a common reference year, OC measurements agreed to within 29-48% and EC measurement to within 20% amongst the different networks. Fitted with a sigmoid function, elevated OC and EC concentrations were found when ambient temperature exceeded 10 °C. These increased ambient concentrations of OC during summer were attributed to the secondary organic aerosol (SOA) formation and forest fire emissions, while elevated EC concentrations were attributed to forest fire emissions and increased vehicle emissions. The observations from this study suggest that carbonaceous aerosol measurements, especially EC, can be synchronized across networks if sample collection and analytical method in each network remain internally consistent. This study allows the generation of regional to continental-scale-harmonized concentration data sets for benchmarking of atmospheric chemical transport models that determine emission sources and sinks, and assess the effectiveness of government mitigation policies in improving air quality and reducing reliance on fossil fuel consumption.

## Introduction

Carbonaceous aerosols, including organic carbon (OC) and elemental carbon (EC; often referred to as black carbon (BC)), make up a large fraction of the atmospheric particulate matter (PM) mass. Atmospheric OC and EC particles that are emitted directly into the atmosphere have both natural (e.g., biomass burning or forest fires) and anthropogenic (e.g., internal combustion engines) sources. A significant amount of the particulate OC is also formed in the atmosphere through oxidation and condensation of volatile organic compounds (e.g., isoprene and terpenes), which are emitted directly from vegetation. BC is a by-product of incomplete combustion of hydrocarbon fuels, generated mainly from fossil fuel combustion and biomass burning. Atmospheric particles have direct and indirect influences on climate, visibility, air quality, ecosystems, and adverse human health effects (Bond et al., 2013; Japar et al 1986; Lesins et al., 2002; Watson, 2002). Atmospheric BC absorbs solar radiation while





OC scatters (Schulz et al., 2006).  However, BC and OC co-exist in atmospheric particles and the net
radiative forcing of the aerosol particles depends on the particle size, composition, and the mixing state
of the particles, while all of these variables also change as aerosol particles age (Fuller et al., 1999; Lesins
et al., 2002).  Long-term atmospheric OC and EC measurements provide necessary benchmark data for
understanding inter-annual trends and seasonal variations and for constraining BC sources (Collaud
Coen et al. 2013).  They are also needed for determining changes of emissions and their impacts on
atmospheric processing and developing/verifying the effectiveness of future environmental and health-
related policies (Chen et al. 2012).

Conducting long-term ambient BC mass measurements is challenging in part due to the lack of a

universally accepted definition of BC.  The scientific community generally accepts the definitions from
Bond et al. (2013) that BC particles possess the following properties: (1) strongly absorbing in the visual
spectrum with an inverse wavelength ($\lambda$) dependence (i.e., $\lambda^{-1}$) (Bond and Bergstrom 2006), (2)
refractory in nature with a vaporization temperature near 4000 K (Schwarz et al., 2006), (3) insoluble in
water and common organic solvents (Fung 1990), (4) aggregate form (Kittelson, 1998), and (5)
chemically inertness in the atmosphere, as graphitic carbon (Bond et al. 2013).  BC is a generic term in
the literature and is often interchanged with other terms such as EC, soot, refractory BC, light absorbing
carbon, or equivalent BC (Petzold et al 2013).  The use of different terminology is linked to the different
methodologies used to measure different physical or chemical properties of BC.  In this study, the term
BC is referred to a substance that absorbs a significant amount of visible light and was created during
incomplete combustion, from either internal combustion engines or biomass burning.  EC is referred to
the carbon mass determined from the thermal evolution analysis (TEA) or thermal optical analysis (TOA)
of carbonaceous materials at the highest temperature set point (e.g., >550 °C) under an oxygenated
environment.  It is also assumed that ambient EC and BC concentrations resemble each other.

TOA and TEA have been applied in many long-term monitoring networks with various protocols

to quantify OC and EC concentrations from aerosol deposits on quartz-fiber filters (Birch and Cary, 1996;
Cachier et al., 1989; Cavalli et al., 2010; Chow et al., 1993; Huang et al., 2006; Huntzicker et al., 1982)
due to the simplicity in filter sample collection and the analytical procedures.  TOA and TEA provide a
direct measurement of carbon mass as part of the gravimetric PM mass.  One of the limitations of TOA
and TEA is the need for sufficient sampling time to accumulate enough mass for precise measurements
(i.e., ensuring a high signal to noise ratio) which constrains the temporal resolution of such samples.  In
addition, EC and OC are defined differently in different protocols and could affect the absolute mass



values measured.  Generally, OC is quantified under a pure helium (He) atmosphere at a low heating
temperature whereas EC is quantified under an oxygen ($O_2$)/He atmosphere at high temperatures.
Estimates of total carbon (TC=OC+EC) derived from TOA and TEA methods are generally consistent,
whereby OC estimates agree within 10-20%, but larger differences are found for EC, owing to its smaller
contribution to TC (Cavalli et al., 2010; Chow et al., 1993; 2001; 2005; Countess 1990; Watson et al.,

2005).

During thermal analysis, some of the OC chars to form pyrolyzed organic carbon (POC) when

heated in the inert He atmosphere, darkening the filter (Chow et al., 2004; Watson et al. 2005).  When
$O_2$ is added, POC combusts with EC and leads to an overestimation of EC in the PM deposit.  The
formation of POC depends on the nature of the organic materials, the amount of the oxygenated
compounds in the collected particles, the rate, duration, and temperature of the heating, and the supply
of $O_2$ in the carrier gas (Cachier et al. 1989; Chan et al., 2010; Han et al. 2007; Yang and Yu, 2002).  POC
in TOA is estimated by monitoring reflectance and/or transmittance of a 633-650 nm laser beam, with
the resulting EC method termed thermal optical reflectance (TOR) or thermal optical transmittance
(TOT).  When the reflected or transmitted laser signal returns to its initial intensity at the start of the
analysis (i.e., at OC/EC split point), it is assumed that artifact POC has left the sample and the remaining
carbon belongs to EC.  The carbon mass before the split point is defined as OC whereas that after the
split point is defined as EC.  POC is defined as the mass determined between the time when $O_2$ is
introduced and the OC/EC split point.

Quartz-fiber filters adsorb organic vapors (Chow et al., 2009; Turpin et al., 1994; Viana et al.,

2006; Watson et al., 2010), resulting in non-PM contributions to OC and charring enhancement within
the filter.  These vapors are adsorbed passively when the filter is exposed to air and more so as air is
drawn through the filter during PM sampling.  Sampling at low filter face velocities for long period of
time could lead to more adsorption (McDow and Huntzicker, 1990), while using high filter face velocities
for longer sample durations may result in evaporation of semi-volatile compounds as negative artifact
(Khalek, 2008; Sutter et al., 2010; Yang et al., 2011).  The positive OC artifact from adsorption usually
exceeds the negative evaporation artifact, especially at low temperatures, resulting in OC
overestimation.  This can be corrected by subtracting the OC concentration from field blanks or backup
filters located downstream of a Teflon-membrane or quartz-fiber filter (Chow et al., 2010; Watson et al.,

2005; 2010).



Previous studies further suggested that TOT could over-estimate the POC mass more than TOR,

resulting in higher POC (and lower EC) because of the charring of the adsorbed organic vapors within the

filter (Chow et al 2004; Countess 1990).  Since only a portion (0.5-1.5 cm$^2$) of the filter is analyzed,

inhomogeneous PM deposits add to measurement uncertainty when OC and EC are normalized to the

entire filter deposit area.  Deposits that are light or too dark can cause unstable laser signals that affect

the OC/EC split (Watson et al., 2005).

This study evaluated the consistency and comparability of co-located carbonaceous aerosol

measurements by three North American networks (IMPROVE, CAPMoN, CABM) over 10 years.  These

networks use different sampling instruments, frequencies, and durations, analytical methods, and

artifact corrections.  The investigation identified potential issues and determined solutions for improving

the compatibility of the different measurements.  When combining all measurements from all networks

at various sites, it offers the possibility to create a regional- to continental-scale, harmonized carbon

concentration dataset, which is important and necessary for constraining model input for understanding

the OC and EC sources.

**Sampling and Measurements**

***Sampling Site***

The sampling station is the Center for Atmospheric Research Experiments (CARE) located near

Egbert, Ontario (44°12´ N, 79°48´ W, 251 m a.s.l.), Canada.  This station is owned and operated by

Environment and Climate Change Canada (ECCC), and is located 70 km NNW of the city of Toronto.

There are no major local anthropogenic sources within about 10 km of the site.  Air that reaches this site

from southern Ontario and the northeastern United States typically carries urban or anthropogenic

combustion pollutants that were emitted within last two days (Rupakheti et al. 2005; Chan and

Mozurkewich 2007; Chan et al., 2010).  Air from the north generally contains biogenic emissions and is

often accompanied with the presence of SOA during summer (Chan et al., 2010; Slowik et al., 2010).

Table 1 compares the instrument and analytical specifications among the three networks.

***The IMPROVE Network***

The IMPROVE network, established in 1987, includes regional-scale monitors for detecting

visibility trends, understanding long-range transport, and evaluating atmospheric processes (Malm

1989; Yu et al. 2004).  IMPROVE operates about 150 sites and provides long-term records of PM$_{10}$ and



PM$_{2.5}$ (particles with aerodynamic diameter less than 10 and 2.5 microns, respectively) mass as well as
PM$_{2.5}$ composition, including anions (i.e., chloride, nitrate, and sulfate), and carbon (OC and EC).
IMPROVE 24-hour samples at Egbert were acquired once every third day from 2006 to 2015.  The
sampling period was from 0800 to 0800 local standard time (LST) except for August 16, 2006 through
October 24, 2008 (from 0000 to 0000 LST).  Module C of the IMPROVE sampler uses a modified air-
industrial hygiene laboratory (AIHL) cyclone with a 2.5 μm cut point at a flow rate of 22.8 liters per
minute (L/min).  PM samples were collected onto a 25 mm diameter quartz-fiber filter (Tissue quartz,
Pall Life Sciences, Ann Arbor, MI, USA), which were pre-fired at 900°C for four hours.  Once sampled,
filters were stored in freezer until they were ready to be analyzed.  All samples were analyzed by the
IMPROVE_A thermal/optical reflectance protocol (Figure S1a; Supplementary information) (Chow et al.,
2007) as shown in Table S1 (Supplementary information).  The IMPROVE measurements (denoted as
IMPROVE_A TOR) were obtained from the website http://vista.cira.colostate.edu/IMPROVE (Malm et
al., 1994; IMPROVE, 2017).
***The CAPMoN Network***

CAPMoN was established in 1983 to understand the source impacts of acid rain-related

pollutants from long-range transport to the Canadian atmosphere and soil.  The network operates 30
regionally representatives sites (as of 2015) across Canada with most located in Ontario and Quebec.
Measurements include PM, trace gases, mercury (both in air and precipitation), tropospheric ozone, and
multiple inorganic ions in air and precipitation (https://www.canada.ca/en/environment-climate-
change/services/air-pollution/monitoring-networks-data/canadian-air-precipitation.html).

Twenty-four-hour samples (0800 to 0800 LST) were acquired every third day from 2005 to 2015

using the Modified Rupprecht and Patashnick (R&P) Model 2300 PM$_{2.5}$ Speciation Sampler with
ChemComb cartridges and PM$_{2.5}$ impactor plates with impactor foam to direct particles onto a 47 mm
diameter tissue quartz-fiber filter operated at 10 L/min (Thermo Scientific, Waltham, MA, USA).  A
second parallel cartridge is configured with a 47 mm front Teflon-membrane filter and a quartz-fiber
backup filter to estimate vapor adsorption artifact.  All quartz-fiber filters were pre-fired at either 800°C
or 900°C for over two hours and cooled at 105°C overnight and stored in freezer (-15 °C) before being
used for sampling.  All sampled filters were also stored in freezer until they are ready for analysis.

Carbon was determined using the Sunset laboratory-based carbon analyzer

(http://www.sunlab.com/) following the IMPROVE-TOT protocol from 2005 to 2007 (denoted as Sunset-



TOT), then by DRI Model 2001 Thermal/Optical Carbon Analyzer following the IMPROVE-TOR protocol
(denoted as DRI-TOR) from 2008 to 2015 (Chow et al., 1993).  As shown in Table S1, the temperature
settings for IMPROVE (i.e., DRI-TOR) protocol for CAPMoN samples are lower than those of IMPROVE_A
protocol for IMPROVE samples by 20°C to 40°C (Figure S1b).  This small difference in the temperature-
ramp between these protocols results in correlated but different OC, EC, and TC mass (Chow et al.,

2007).

***The ECCC Canadian Aerosol Baseline Network***
The Climate Chemistry Measurements and Research (CCMR) Section in the Climate Research
Division of ECCC has operated the Canadian Aerosol Baseline Measurement (CABM) network since 2005
to acquire data relevant to climate change (https://www.canada.ca/en/environment-climate-
change/services/climate-change/science-research-data/greenhouse-gases-aerosols-
monitoring/canadian-aerosol-baseline-measurement-program.html).  The CABM network includes 6
sites (as of 2016) for aerosol chemical, physical, and optical measurements that cover ecosystems at
costal, interior urban/rural areas, boreal forests, and the Arctic.  Measurements are intended to
elucidate influences from various emission sources on regional background air, including biogenic
emissions, biomass burning as well as anthropogenic contributions from industrial/urban areas.
The CABM filter pack system uses a $PM_{2.5}$ stainless steel cyclone (URG-2000-30EHS) operated at
16.7 L/min for sampling from 2006 to 2015 with an operator manually changing the 47 mm quartz-fiber
filter on a weekly basis.  All quartz-fiber filters were pre-fired at 900°C overnight prior being sampled.
Once sampled, filters were stored in freezer until they were ready to be analyzed.  A TEA method,
EnCan-Total-900 (ECT9), developed by Huang et al. (2006) and refined later (Chan et al., 2010), was used
to analyze the OC, POC, and EC on the quartz-fiber filters using a Sunset laboratory-based carbon
analyzer.  The ECT9 protocol was developed to permit stable carbon isotope ($^{13}$C) analysis of the OC and
EC masses without causing isotope fractionation, as it was demonstrated by Huang et al. (2006).  This
method first heats the filter at 550°C and 870°C for 600 s each in the He atmosphere to determine OC
and POC (including carbonate carbon; CC), respectively, and then combusts the sample at 900°C under
2% $O_2$ and 98% He atmosphere for 420 s to determine EC (Figure S1c and Table S1).  The ECT9 POC
definition (released as $CO_2$ at 870 °C) includes the charred OC, and some calcium carbonate ($CaCO_3$) that
decomposes at 830°C, as well as any refractory OC that is not combusted at 550°C.  Chan et al. (2010)
found that POC determined by ECT9 was proportional to the oxygenated compounds (e.g., aged aerosol
from atmospheric photochemical reaction) and possibly humic-like materials.  Consistent with the





IMPROVE_A TOR protocol (Chow et al., 2007), OC is defined as the sum of OC and POC, as CC is usually
negligible in PM$_{2.5}$.

CABM sites are also equipped with Particle Soot Absorption Photometer (PSAP; Radiance

Research, Seattle, WA, USA) that monitor aerosol light absorption as changes in the amount of light
transmitted through a quartz-fiber filter.  Assuming the mass absorption coefficient (MAC) for aerosol is
constant at Egbert, the one minute PSAP absorption measurements is linearly proportional to the BC or
EC oncentrations.
***Differences in Sampling and Analysis among Networks***

Depending on the sharpness (i.e., slope) of the inlet sampling effectiveness curve (Watson et al.,

1983), different size-selective inlets may introduce measurement uncertainties.  CAPMoN uses an
impactor whereas CABM and IMPROVE use cyclones.  Impactor may have larger pressure drops across
the inlet that might enhance semi-volatile PM evaporation.  Larger solid particles might bounce when in
contact with the impactor and be re-entrained in the PM$_{2.5}$ samples if the impactor is overloaded (Flagan
and Seinfeld, 1998; Hinds, 1999).  Atmospheric mass size distributions typically peak at about 10 μm
with a minimum near 2.5 μm, therefore, the difference in mass collected with different impactors or
cyclones among the three networks is not expected to be large (Watson and Chow, 2011).

The small filter disc (25 mm diameter) and high flow rate (22.8 L/min) in the IMPROVE sampler

result in a 5- to 7-fold higher filter face velocity (i.e., 107.7 versus 16-20 cm/s) than that for the CAPMoN
and CABM samplers.  McDow and Huntzicker (1990) assert that higher filter face velocity may reduce
sampling artifacts.  However, very high face velocity (>100 cm/s) may enhance OC volatilization (Khalek

2008).

Both CAPMoN and IMPROVE networks correct for vapor adsorption, while CABM network does

not.  CAPMoN subtracts the organic artifact from the parallel channel, whereas IMPROVE (up until 2015)
used monthly median OC values obtained from the backup quartz-filters from 13 sites (not including
Egbert).

Multiple studies show that using the same TOA protocol on both DRI and Sunset carbon

analyzers can produce comparable TC concentrations.  However, large differences in EC are found
between the reflectance and transmittance POC correction (Chow et al., 2004; 2005; Watson et al.,
2005).  Difference in OC and EC definitions among different TOA and TEA protocols introduce



measurement uncertainties.  Among the TOA methods, how POC is determined from the laser signals at
different temperatures in the inert He atmosphere introduce uncertainties.  Large uncertainties in laser
transmittance were found for lightly- and heavily-loaded samples (Birch and Cary, 1996).  For the CABM
samples, the POC determined at 870 °C by ECT9 represents different OC properties and does not equal
the charred OC obtained by Sunset-TOT, DRI-TOR, or IMPROVE_A TOR.
**NIST urban dust standard comparison (SRM 8785 & 1649a)**

The consistency of the OCEC measurements obtained between the ECT9 and the IMPROVE_A

method was assessed by measuring four replicates of the National Institute of Standards and
Technology (NIST) Urban Dust Standard Reference Material (SRM) 8785 (Cavanagh and Watters, 2005;
Klouda et al., 2005).  These samples were produced by resuspension of the original SRM 1649a urban
dust sample, followed by collection of the fine fraction ($PM_{2.5}$) on quartz-fiber filters (Klouda et al., 2005;
May and Trahey, 2001).  Past studies on SRM 1649a and SRM 8785 have shown consistent composition
(Currie et al., 2002; Klouda et al., 2005).  The SRM 8785 filters with mass loading of 624-2262 μg were
analyzed following the ECT9 method by the ECCC laboratory and the IMPROVE_A protocol by the DRI
laboratory during 2009-2010.

Figure 1 shows reasonable correlations with 21-25% higher TC and EC by the ECT9 method.  The

values in the SRM 8785 certificate were reported as grams of OC or EC per grams of PM mass, which are
average mass ratios based on analysis of a small numbers of randomly selected samples.  Figure 2 shows
that IMPROVE_A protocol by DRI compared well with the certificate values.  Ratios measured with ECT9
were greater, but not significantly different from the certificate values.

The parameter EC/TC, calculated based on the reported certificate values, were compared with

the average EC/TC values determined from the inter-comparison study (ICP) by the DRI group (using
IMPROVE_A) and the ECCC group (using ECT9) (Fig. 2d).  These results show that EC/TC reported by both
DRI and CCMR were statistically the same as the certificate value.

Finally, the EC/TC value was further verified by analyzing multiple SRM 1649a sample with the

ECT9 method.  The combusted $CO_2$ from OC, EC, and TC were analyzed for the isotope ratios (i.e.,
$^{14}C/^{12}C$) expressed as a fraction of modern carbon (i.e., $FM_i$ is the ratio of $^{14}C/^{12}C$ in the sample i, relative
to a modern carbon standard) for individual mass fractions (i.e., $FM_{TC}$, $FM_{OC}$, and $FM_{EC}$).  Using isotopic
mass balance, the EC/TC ratio can be derived from Eq. [1]:





$$FM_{TC} = FM_{OC} \times \left(1 - \frac{EC}{TC}\right) + FM_{EC} \times \frac{EC}{TC} \qquad [1]$$


The $^{14}C/^{12}C$ abundances were determined by off-line combustion method at the Keck Carbon Cycle
accelerator mass spectrometry (KCCAMS) Facility at University of California Irvine. A $FM_{TC}$ value of 0.512
was obtained, which is close to certificate values that range from 0.505 to 0.61 (Currie et al., 2002).
Average measured values of $FM_{OC}$ and $FM_{EC}$ for the SRM 1649a via ECT9 were 0.634 (n=3) and 0.349
(n=3), respectively. This yields an EC/TC ratio of 0.425, close to the reported certificate value of 0.49
and the IMPROVE_A value of 0.47 (Figure 2d), reconfirming a good separation of OC from EC using the
ECT9 method.
**Results and Discussion**
***Weekly versus 24-hour Samples***
Comparing two years of Egbert measurements (2005-2007), Yang et al. (2011) suggested that
integrated weekly samples may experience reduced vapor adsorption but increased losses of semi-
volatile organics leading to lower OC measurements. Weekly EC values were higher than those from 24-
hour samples, which were attributed to the higher analytical uncertainties for the lower loadings on the
24-hr samples (Yang et al., 2011). Here, the effect of different sample duration on EC concentrations is
assessed using five years (2010-2015) of real-time (1 min average) PSAP particle light absorption
measurements (at 567 nm). The results demonstrate that both data sets capture the variations
adequately (Figure 3a) with correlated monthly averages (Figure 3b). Figure 3c shows a good correlation
(r=0.78) between the weekly and every third day 24-hour samples with a slope of 0.96. Assuming the
variations in light absorption can represent the variations in EC, these results suggest that monthly
averaged EC based on weekly sampling is about 4% lower.
***Vapor Adsorption Corrections***
Figure 4 shows the carbon concentration time series with and without the artifact correction for
CAPMoN samples. Vapor adsorption contributes to a large amount of the measured OC (Figure 4a), but
a negligibly amount to EC (Figure 4b) and POC after 2008 (Figure 4c). The median vapor adsorption
artifact was 0.79 µg/m$^3$ from 2008 to 2015 for DRI-TOR, representing about 50.9% of the uncorrected
OC, compared to 0.92 µg/m$^3$ (43.3% of uncorrected OC) using the Sunset-TOT before 2008
(Supplemental Figure S2). Least square regressions between corrected and uncorrected carbon in
Figure 5 shows a slope of 0.52 for OC and 0.56 for TC with good correlations (r=0.93-0.94). Sunset-TOT



measurements acquired prior 2008 are mostly scattered around the regression line, with higher
concentrations. On average, about 48% of the uncorrected OC (0.84 µg/m³) can be attributed to vapor
adsorption. The low filter face velocity (15.5 cm/s) in CAPMoN samples could be one of the contributing
factors.
Figure 5c indicates that EC concentrations are 7.8% (0.02 µg/m³) lower after artifact correction.
These levels are close to the detection limit of 0.022 µg/m³ and within analytical uncertainties (Chow et
al., 1993). Some Sunset-TOT EC measurements are scattered from the regression line, indicating a more
accurate and consistent adsorption correction for DRI-TOR (Figure 5b). Although not expected to impact
EC concentration, vapor adsorption directly affects POC correction and thus influences EC mass
determination.
Figure 5d shows that 4.3% (0.01 µg/m³) of POC was caused by vapor adsorption using the DRI-
TOR protocol. For Sunset-TOT (red open circles), however, up to 21.1% (0.17 µgC/m³) of the POC was
detected on the backup filter. Filter transmittance is influenced by both surface and within filter
charring and EC from different sources have been observed to have different filter penetration depths
(Chen et al., 2004; Chow et al., 2004), an optical correction by reflectance appears to be more
appropriate when POC/EC ratio in measurements are high. Regardless, the absolute POC and EC
concentrations were much lower than OC and the adsorption correction on TC is mostly attributed to
the OC artifact.
Since the CAPMoN aerosol deposits were acquired at a low filter face velocity (15.5 cm/s), it is
expected that the magnitude of the vapor adsorption correction would be smaller for the IMPROVE
samples due to the use of higher filter face velocity. This is supported by the observations from Watson
et al. (2009) at six anchor IMPROVE sites, suggesting that vapor adsorption obtained from backup quartz
filters represented about 23% of the uncorrected OC values, whereas those obtained from field blanks
were averaged to be about 18%. In comparison with the IMPROVE measurements at Egbert, the vapor
adsorption obtained from field blank represent about 16% (or 0.18 µg/m³) of the uncorrected OC
measurements. Filter fibers are saturated over a long sampling interval (Khalek, 2008; Watson et al.,
2009), thus, artifacts for the CABM samples are expected to be lower relatively.
**CAPMoN vs. IMPROVE Measurements**
Temporal variations of DRI-TOR CAPMoN measurements are comparable to the IMPROVE_A
TOR protocol (Figure 6), showing a similar temporal pattern with elevated peaks found in mid-summer



(July).  High correlations are found for DRI-TOR OC and TC with IMPROVE_A measurements (r=0.90-0.91;
Table 2) while lower correlations (r=0.78-0.79) are seen for Sunset-TOT data.  Good correlations are
observed between the DRI-TOR and IMPROVE_A TOR POC measurements (r=0.85) but much lower
correlations are observed for Sunset-TOT and IMPROVE_A POC measurements (r=0.70).  Correlations
between DRI-TOR EC and IMPROVE_A TOR EC are high (r=0.81) but it is not the case between
Sunset-TOT EC and IMPROVE_A TOR EC (r=0.33).

When fitting DRI-TOR and Sunset-TOT measurements to IMPROVE_A TOR measurements

through the origin (i.e., Regression 1) typically yields less than unity slopes (0.64-0.97; Table 2),
suggesting that the carbonaceous masses reported by CAPMoN were in general lower than IMPROVE.
Fitting the measurements allowing an intercept (i.e., Regression 2) typically yields least square slopes
close to unity (>0.92) with small intercepts.

The effect of using transmittance or reflectance for POC determination is apparent.  The

Sunset-TOT POC correction is larger because transmittance is affected by the charred OC within the
filter.  This is consistent with the larger regression slopes in POC (Regression 1: 1.8) between Sunset-TOT
and IMPROVE_A TOR protocol than the slope in POC (1.0) between the DRI-TOR and IMPROVE_A TOR
protocol.
**CABM vs. IMPROVE Measurements**

Figure 6 shows the temporal variations of the ECT9 CABM measurements with other networks.

The temporal variations of the CABM measurements were consistent with the temporal trends of
measurements from the other two networks.  While ECT9 OC concentrations are comparable (±~15%)
with the IMPROVE_A TOR measurements, higher TC and EC concentrations are found in CABM samples.

The ECT9 versus IMPROVE_A Regression 1 slopes are equal to or greater than unity, ranging

from 1.2 to 1.8 (Table 2).  Linear regression with intercept (i.e., Regression 2) yields lower slopes
(0.6-1.7) with positive intercepts (0.06-0.18 $\mu$g/m$^3$), signifying higher TC and EC concentrations for ECT9
samples.  Higher intercepts (0.12-0.18 $\mu$g/m$^3$) for TC, OC, and POC are consistent with ECT9
measurements uncorrected for vapor adsorption.  The ECT9 method yielded 66-83% higher EC than
IMPROVE_A TOR, with moderate correlation (r=0.74).  Differences in combustion temperatures, OC/EC
split, inhomogeneous deposition of mass loading on filter spots could contribute to these discrepancies.
Heating under an oxidative environment at a constant temperature of 900 °C in the ECT9 protocol could



combust more highly refractory carbon than the IMPROVE_A protocol, which only heats progressively
from 580 °C to 840 °C.  When plotted on different scales, Figure S3 shows that the two EC data sets track
well, capturing both long-term trends and short-term variations.

A slope approaching unity (1.00) was obtained when fitting the ECT9 POC to IMPROVE_A TOR

POC through the origin (Figure 7d).  Refitting the data allowing an intercept leads to a slope of 0.62 with
a y-intercept (0.12; Table 2), comparable in magnitude to the vapor adsorption artifact.  The correlation
coefficient between ECT9 POC and IMPOVE_A TOR POC is low (r=0.46; Table 3).  However, there is a
significant correlation found between the IMPROVE_A TOR POC and IMPROVE_A TOR OC (r=0.91), and
even to a lesser extent between IMPROVE_A TOR POC and IMPROVE_A TOR EC (r=0.71).  In comparison,
ECT9 POC has weak correlation with ECT9 OC (r=0.65) and ECT9 EC (r=0.37).
**Comparison of the Normalized Time Series**

Measurements from the networks are normalized in Figure 8 to assess the comparability as

follows:

$[x]'_t = [x]_t / [x]_{to}$                              [2]

Where $[x]'_t$ is the normalized concentration at time $t$; $[x]_t$ and $[x]_{to}$ are monthly carbonaceous aerosol
mass concentrations at time $t$ and in January 2008, respectively.  Figure 8 also includes the monthly
average temperatures (green dotted curve), wind speed and direction (in the form of wind barb).

Normalized TC from the three networks correlate well (r≥0.79; Table 3).  Figure 8a shows that

normalized TC values tracked each other and represented the annual pattern, with DRI-TOR showing the
largest TC, followed by IMPROVE_A TOR and ECT9.  TC concentrations peaked around each July which
experienced the highest temperatures.

Comparable temporal variations among these networks are also found for EC (Figure 8c) with

good correlations (r=0.69-0.81).  Figure 9e shows that DRI-TOR and IMPROVE_A TOR EC tracked well
(slope of 1.03±0.02) with greater scatter for the ECT9 samples (slope=1.125) during summer (May-Oct in
Figure 9f; defined based on ambient temperature).

Temporal variations of the normalized OC (Figure 8b) are similar to TC variations with high

correlations (r=0.86-0.90; Table 3).  However, large variations were found in Regression slopes from 0.71
to 1.32 (Table 4).



Figures 9c and 9d show that OC measurements were higher during summer (May-October).

Monthly wind roses (Figures S4) show prevailing winds from the NW during summer.  Air masses
reaching Egbert during summer represent a mixture of the anthropogenic emissions from the south and
the biogenic emissions from the north.  Intense solar radiation received during summer could result in
increased SOA formation.  The limited solar radiation during winter hindered the SOA formation
resulting in less data scatter.  The extent of organic vapor desorption may be enhanced for the week-
long samples in CABM network during summer.

The largest variations are found for POC (Figure 8d) with the lowest normalized POC

concentrations found for ECT9 samples.  Table 3 shows that POC concentrations are correlated with OC
for both IMPROVE_A TOR and DRI-TOR samples (r=0.91-0.92) but not for ECT9 samples (r=0.65).  POC
from DRI-TOR and IMPROVE_A TOR samples also correlated (r=0.85), but not with ECT9 samples (r=0.44-
0.46).  These observations shows that the POC definition in ECT9 is not just a charred OC correction but
likely include the characterization of other oxygenated organic materials (Chan et al., 2010).  Additional
research is needed to verify if the increased SOA formation during the summer season could increase
the variation of the ECT9 POC with POC defined by other methods.
**Seasonality in Carbon**

A sigmoid function was applied to characterize the relationship between carbon and ambient

temperature.  The Sigmoid function has a characteristic "S" shape and represents an integral of a
Gaussian function.  Figure 9 shows elevated carbon during summer, consistent with the observations
from Yang et al. (2011) and Healy et al. (2017).  Relationships between carbon concentrations and
ambient temperatures are shown in Figure S5.  Apparent increases in OC and TC concentrations are
found when ambient temperatures exceed about 10 °C; a phenomenon not as apparent in EC.  EC from
the week-long CABM samples are more scattered.

The TC, OC, and EC from all measurements are averaged and shown in Figure 10 with the

following best-fitted sigmoid functions:
$$TC = 1.053 + \left\{ \frac{3.558}{1+exp\left(\frac{23.081-T}{3.760}\right)} \right\} \qquad [3]$$

$$OC = 0.780 + \left\{ \frac{1.838}{1+exp\left(\frac{20.089-T}{2.978}\right)} \right\} \qquad [4]$$



$$EC = 0.239 + \left\{ \frac{1.446}{1 + exp\left( \frac{34.776 - T}{8.404} \right)} \right\}$$
[5]

Equations [3]-[5] show that lower limits of the observed TC, OC, and EC concentrations are 1.05, 0.78,
and 0.24 µgC/m$^3$, with the half way of the maximum growth curve occurring at about 23 °C, 20 °C, and
35 °C, respectively.  The predicted maximum concentrations for TC, OC, and EC are 4.61, 2.62, and 1.69
µgC/m$^3$, respectively.
To determine the air mass origins, a Lagrangian particle dispersion transport model (FLEXible
PARTicle dispersion model; FLEXPART) (Stohl et al., 2005) was applied to obtain daily five-day back-
trajectories from Egbert from 2006 to 2015.  Figure S6 summarizes the average FLEXPART footprints for
summer (May-Oct) and winter (Nov-Apr) seasons, showing the probability of air masses originating from
various regions.  These results indicate regional contributions from boreal forest in the northern part of
Ontario and Quebec, as well as anthropogenic emissions from the northern U.S.  Five-day trajectories
show larger concentrations from the N and NW, consistent with wind roses shown in Figure S4.
At low ambient temperatures, primary emissions (e.g., local transportation, residual heating,
and industrial activities) account for most of the ambient OC and EC (Ding et al., 2014).  Increased
human activities (e.g., traveling by car and barbecuing) during warmer weather could lead to increased
emissions.  High ambient temperature also leads to increased biogenic emissions (e.g., monoterpenes)
from the boreal forest and increased SOA formation (Chan et al., 2010; Leaitch et al., 2011; Passonen et
al., 2013; Tunved et al., 2006).  The central and eastern boreal forest fire season typically occurs from
May to August when ambient air is dry and hot, resulting in generally increased OC and EC emissions
(Lavoué et al 2000).  Transboundary transport of biomass burning emissions from the U.S. could also
contribute to the higher concentrations in southern Ontario (Healy et al. 2017).  Increasing ambient
temperature from 10 °C to 20 °C leads to higher OC concentrations from 0.84 to 1.61 µgC/m$^3$ (91.7%
increase) and EC concentration from 0.31 to 0.45 µgC/m$^3$ (45.2% increase).  The temperature
dependency of OC and EC suggests a potential climate feedback mechanism consistent with the
observations from Leaitch at al. (2011) and Passonen et al. (2013).
Chan et al. (2010) showed that ECT9 POC possesses a positive relationship with oxygenated
organics and aged aerosol particles.  The seasonality in ECT9 POC is compared with the average OC and
EC seasonality observed at Egbert (Figure 10d).  Interestingly, the ECT9 POC concentration does not
show a gradual exponential shape of function as for OC and EC (Figures 10b and 10c).  Instead, it shows



a small but obvious two-step function when plotted against ambient temperature.  The ECT9 POC results
(Figure 10d) suggest constant sources of background emissions of possible oxygenated organic
compounds with additional secondary formation at higher temperatures (e.g., >15 °C).  Future study is
needed to verify this.

**Conclusions**

Ten years of OC and EC measurements at Egbert were obtained from three independent

networks (IMPROVE, CAPMoN, CABM).  Differences in carbon concentrations were attributed to
different sampling methods, analytical protocols, and filter artifact corrections.  Vapor adsorption did
not affect EC values but contributed about 48% (or 0.84 µg/m$^3$) of the measured OC for the CAPMoN
network with the lowest filter face velocity of 15.5 cm/s.  When sampling at a filter face velocity of 108
cm/s, the IMPROVE field blanks account for about 16% (or 0.18 µg/m$^3$) of the measured OC.  TC
measurements differences were influenced by the uncorrected vapor adsorption artifact as a result of
the OC artifact as in the CABM measurements.  Pyrolyzed OC (POC) from both DRI-TOR (i.e., IMPROVE)
and IMPROVE_A TOR protocols correlated with OC (r=0.91-0.92), indicative of the charring property of
the measured OC.  ECT9 POC was only weakly correlated with OC (r=0.65) and had weak correlations
with POC from DRI-TOR and IMPROVE_A TOR (r=0.44 and 0.46), suggesting ECT9 POC includes
compounds with different properties under high temperature gasification (870 °C).

Analyzing the SRM 8785 standard reference samples demonstrate the consistency in long-term

carbon measurements.  Although the inter-comparison of SRM 8785 samples showed 20-25% higher TC
and EC by ECT9 method, no statistical difference was found for the EC/TC ratios, which was further
supported by isotope measurements.  The normalized measurements obtained from different networks
agree within 48% for OC and 20% for EC, suggesting measurements from different networks can be
synchronized to generate harmonized carbonaceous concentration maps for constraining/verifying
climate model and for investigating emission sources and sinks.  A North America harmonized
carbonaceous concentration map is useful for evaluating the national and inter-national mitigation
policies efforts on improving air quality and reducing fossil fuel consumption.

Carbon concentrations exhibited a non-linear positive dependency with ambient temperature,

and this relationship is characterized by a sigmoid function.  Preliminary observations suggested that the
increases in OC and TC concentrations when ambient temperatures rose beyond 10 °C during summer,
is likely corresponding to the sum of SOA formation, influences of forest fires, and increased



anthropogenic activities. The moderate increase in EC concentration with increasing ambient
temperature is believed to be a result of increased primary emissions from anthropogenic activities. The
increase in OC concentration with temperature is consistent with the climate feedback mechanisms
reported from various studies.
**Nomenclature**

| | | |
|---|---|---|
| 469 | AIHL | Air-industrial hygiene laboratory |
| 470 | AMS | Accelerator mass spectrometry |
| 471 | BC | Black carbon |
| 472 | CABM | Canadian Aerosol Baseline Measurement |
| 473 | CAPMoN | Canadian Air and Precipitation Monitoring Network |
| 474 | CARE | Center for Atmospheric Research Experiment |
| 475 | CCMR | Climate Chemistry Measurements and Research |
| 476 | DRI | Desert Research Institute |
| 477 | DRI-TOR | CAPMoN measurements using IMPROVE on DRI analyzer with TOR correction |
| 478 | EC | Elemental carbon |
| 479 | ECCC | Environment and Climate Change Canada |
| 480 | ECT9 | EnCan-Total-900 protocol |
| 481 | FID | Flame ionization detector |
| 482 | FLEXPART | FLEXible PARTicle dispersion model |
| 483 | ICP | Inter-comparison study |
| 484 | IMPROVE | Interagency Monitoring PROtected Visual Environments |
| 485 | IMPROVE_A TOR | IMPROVE_A TOR protocol on DRI analyzer |
| 486 | KCCAMS | Keck Carbon Cycle accelerator mass spectrometry |
| 487 | MAC | Mass absorption coefficient |
| 488 | NIST | National Institute of Standard and Technology |
| 489 | OC | Organic carbon |
| 490 | PM | Particulate matter |
| 491 | POC | Pyrolyzed organic carbon |
| 492 | PSAP | Particle Soot Absorption Photometer |
| 493 | SOA | Secondary organic aerosol |
| 494 | SRM | Standard Reference Material |
| 495 | Sunset-TOT | IMPROVE TOT protocol on Sunset analyzer |
| 496 | TC | Total carbon |
| 497 | TEA | Thermal evolution analysis |
| 498 | TOA | Thermal optical analysis |
| 499 | TOR | Thermal optical reflectance |
| 500 | TOT | Thermal optical transmittance |
| 501 | UCI | University of California Irvine |
| 502 | | |
| 503 | | |



**Acknowledgements**

Authors would like to acknowledge Elton Chan and Douglas Chan of ECCC for providing the FLEXPART
model results and providing technical advice.  IMPROVE measurements were obtained directly from the
IMPROVE web site (http://vista.cira.colostate.edu/IMPROVE/Data/QA_QC/Advisory.htm).  IMPROVE is a
collaborative association of state, tribal, and federal agencies, and international partners.  U.S.
Environmental Protection Agency is the primary funding source, with contracting and research support
from the National Park Service.  IMPROVE carbon analysis was provided by Desert Research Institute.
Funding of this study was initiated by Climate Change Technology and Innovation Initiative (CCTI)
program, operated through Natural Resources Canada (NRCan), and supported by Clean Air Regulatory
Agenda (CARA) initiative and ECCC internal federal funding.

**Supplementary Information:**

Additional details on thermal/optical analysis, the experimental parameters used in different
temperature protocols (IMPROVE, IMPROVE_A, ECT9), radiocarbon analysis, vapor adsorption
uncertainty, seasonality of carbonaceous measurements, wind rose analysis, and FLEXPART back
trajectory analysis at Egbert are included in the Supplementary Information.

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


**Table 1** Specifications for the filter sampling systems used by the three networks.

|  | IMPROVE | CAPMoN | | CABM |
| --- | --- | --- | --- | --- |
| **Data coverage period** | 2005-2015 | 2005-2007 | 2008-2015 | 2005-2015 |
| **Analytical instrument** | DRI | Sunset | DRI | Sunset |
| **Thermal/optical protocol** | IMPROVE_A | IMPROVE | IMPROVE | ECT9 |
| **Pyrolyzed organic carbon detection** | Reflect. | Transmit. | Reflect. & Transmit. | Retention time |
| **Particle size selection method** | Cyclone | Impactor plates | Impactor plates | Cyclone |
| **Particle size cut off diameter (nm)** | 2.5 | 2.5 | 2.5 | 2.5 |
| **Sampling flow rate (L/min)** | 22.8 | 10.0 | 10.0 | 16.7 |
| **Filter media model** | 2500QAT-UP | 2500QAT-UP | 2500QAT-UP | 2500QAT-UP |
| **Quartz filter diameter (mm)** | 25 | 47 | 47 | 47 |
| **Filter deposition exposure area (cm²)** | 3.53 | 10.75 | 10.75 | 13.85 |
| **Filter face velocity (cm/s)** | 107.65 | 15.50 | 15.50 | 20.09 |
| **Sampling frequency** | Daily every 3 days | Daily every 3 days | Daily every 3 days | Integrated weekly |
| **Daily sampled air volume (L/day)** | 31680 | 14400 | 14400 | 24048 |
| **Air volume per sample (m³)** | 31.68 | 14.4 | 14.4 | 168.3 |
| **Positive artifact correction** | Yes | Yes | Yes | No |
| **Filter blank correction** | Yes | No | No | Yes |





**Table 2** Regression results (slope, correlation coefficient, and total number of points) obtained when fitting various CABM (ECT9) and CAPMoN
(Sunset-TOT & DRI-TOR) carbonaceous mass concentration time series against IMPROVE (IMPROVE_A TOR) measurements. IMPROVE_A TOR
and ECT9 measurements cover the period from 2005 to 2015. Sunset-TOT and DRI-TOR measurements cover the periods for 2005-2008 and
2008-2015, respectively. Regression 1 indicates the best-fitted slope through the origin. Regression 2 is the best-fitted slope with intercept (in
brackets).

|  | Regression 1 | Regression 2 | R | N |
|---|---|---|---|---|
| **Sunset-TOT TC vs IMPROVE_A TOR TC** | 0.888±0.033 | 0.713±0.112 (0.301±0.186) | 0.78 | 28 |
| **Sunset-TOT OC vs IMPROVE_A TOR OC** | 0.967±0.041 | 0.873±0.135 (0.125±0.170) | 0.79 | 28 |
| **Sunset-TOT EC vs IMPROVE_A TOR EC** | 0.639±0.042 | 0.233±0.130 (0.171±0.053) | 0.33 | 28 |
| **Sunset-TOT POC vs IMPROVE_A TOR POC** | 1.769±0.091 | 1.776±0.351 (-0.003±0.127) | 0.70 | 28 |
| **DRI-TOR TC vs IMPROVE_A TOR TC** | 0.832±0.015 | 0.946±0.044 (-0.164±0.059) | 0.91 | 93 |
| **DRI-TOR OC vs IMPROVE_A TOR OC** | 0.835±0.017 | 0.934±0.046 (-0.116±0.050) | 0.90 | 93 |
| **DRI-TOR EC vs IMPROVE_A TOR EC** | 0.818±0.019 | 0.929±0.072 (-0.032±0.020) | 0.81 | 93 |
| **DRI-TOR POC vs IMPROVE_A TOR POC** | 0.986±0.028 | 1.230±0.080 (-0.073±0.023) | 0.85 | 93 |
| **ECT9 TC vs IMPROVE_A TOR TC** | 1.304±0.022 | 1.197±0.065 (0.164±0.093) | 0.88 | 107 |
| **ECT9 OC vs IMPROVE_A TOR OC** | 1.149±0.021 | 1.004±0.056 (0.179±0.064) | 0.87 | 107 |
| **ECT9 EC vs IMPROVE_A TOR EC** | 1.834±0.046 | 1.661±0.149 (0.056±0.046) | 0.74 | 107 |
| **ECT9 POC vs IMPROVE_A TOR POC** | 0.998±0.031 | 0.615±0.082 (0.124±0.025) | 0.59 | 107 |






**Table 3** Correlation coefficients (r) of various *normalized* carbonaceous mass measurements among different networks (IMPROVE, CAPMoN and
CABM). All measurements cover the period from 2008 to 2015.

|  |  | IMPROVE_A TOR | | | | DRI-TOR | | | | ECT9 | | | |
| --- | --- | --- | --- | --- | --- | --- | --- | --- | --- | --- | --- | --- | --- |
|  |  | TC | OC | EC | POC | TC | OC | EC | POC | TC | OC | EC | POC |
| IMPROVE_A TOR | TC | 1 | 0.99 | 0.80 | 0.91 | 0.91 | 0.91 | 0.68 | 0.87 | 0.86 | 0.86 | 0.77 | 0.50 |
|  | OC |  | 1 | 0.73 | 0.91 | 0.90 | 0.90 | 0.63 | 0.87 | 0.86 | 0.86 | 0.74 | 0.50 |
|  | EC |  |  | 1 | 0.71 | 0.76 | 0.70 | 0.81 | 0.69 | 0.71 | 0.65 | 0.75 | 0.37 |
|  | POC |  |  |  | 1 | 0.82 | 0.81 | 0.62 | 0.85 | 0.82 | 0.79 | 0.77 | 0.46 |
| DRI-TOR | TC |  |  |  |  | 1 | 0.99 | 0.74 | 0.92 | 0.79 | 0.78 | 0.71 | 0.41 |
|  | OC |  |  |  |  |  | 1 | 0.64 | 0.92 | 0.77 | 0.77 | 0.67 | 0.40 |
|  | EC |  |  |  |  |  |  | 1 | 0.63 | 0.63 | 0.56 | 0.69 | 0.31 |
|  | POC |  |  |  |  |  |  |  | 1 | 0.77 | 0.75 | 0.70 | 0.44 |
| ECT9 | TC |  |  |  |  |  |  |  |  | 1 | 0.98 | 0.91 | 0.58 |
|  | OC |  |  |  |  |  |  |  |  |  | 1 | 0.82 | 0.65 |
|  | EC |  |  |  |  |  |  |  |  |  |  | 1 | 0.37 |
|  | POC |  |  |  |  |  |  |  |  |  |  |  | 1 |


**Table 4** Regression results (slope, correlation coefficient, and total number of points) obtained when fitting various CABM (ECT9)and CAPMoN
(Sunset-TOT & DRI-TOR) *normalized* carbonaceous mass concentration time series against IMPROVE (IMPROVE_A TOR) measurements. All
measurement sets cover the period from 2008 to 2015. Regression 1 indicates the best-fitted slope through the origin. Regression 2 is the best-
fitted slope with intercept (in brackets).

|  | Regression 1 | Regression 2 | R | N |
| --- | --- | --- | --- | --- |
| **DRI-TOR TC vs IMPROVE_A TOR TC** | 1.238±0.022 | 1.408±0.065 (-0.192±0.069) | 0.91 | 93 |
| **DRI-TOR OC vs IMPROVE_A TOR OC** | 1.322±0.027 | 1.479±0.073 (-0.193±0.084) | 0.90 | 93 |
| **DRI-TOR EC vs IMPROVE_A TOR EC** | 1.033±0.024 | 1.173±0.090 (-0.126±0.078) | 0.81 | 93 |
| **DRI-TOR POC vs IMPROVE_A TOR POC** | 1.280±0.036 | 1.597±0.104 (-0.296±0.092) | 0.85 | 93 |
| **ECT9 TC vs IMPROVE_A TOR TC** | 0.804±0.016 | 0.7136±0.045 (0.104±0.049) | 0.86 | 86 |
| **ECT9 OC vs IMPROVE_A TOR OC** | 0.711±0.015 | 0.591±0.038 (0.151±0.044) | 0.86 | 86 |
| **ECT9 EC vs IMPROVE_A TOR EC** | 1.125±0.030 | 1.198±0.114 (-0.066±0.100) | 0.75 | 86 |
| **ECT9 POC vs IMPROVE_A TOR POC** | 0.556±0.020 | 0.221±0.047 (0.318±0.042) | 0.46 | 86 |




**Figure 1** Comparison of: (a) TC, (b) OC, and (c) EC concentrations obtained from the same NIST SRM
8785 filters reported by ECCC following the TEA (ECT9) method and by DRI following the IMPROVE_A
protocol during the inter-comparison study in 2009/2010.

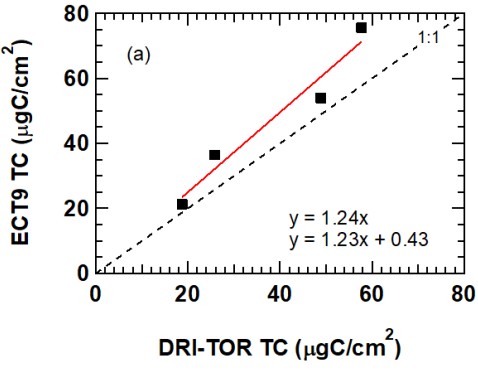
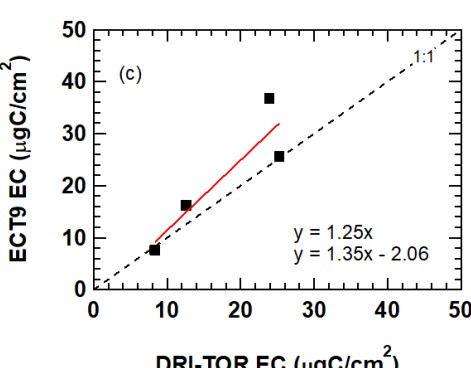


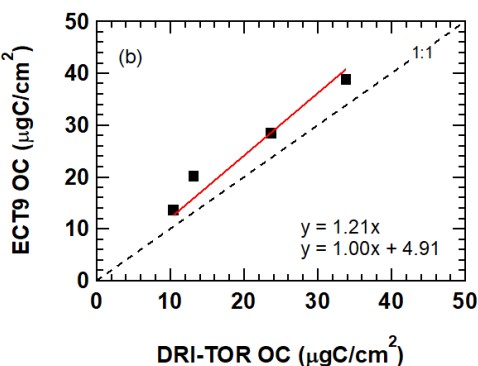








**Figure 2** Comparison of the TC, OC, and EC measurements of the NIST SRM samples reported by the
ECCC and DRI groups during the inter-comparison study conducted between 2009 and 2010.
"Reported" represent the published value in the NIST SRM certificate (Cavanagh and Watters, 2005).
Error bars represent uncertainties covering 95% confidence interval. In (d), the ECT9 value (in green)
represents the calculated EC/TC ratio determined based on stable carbon isotope measurement
obtained from the SRM 1649a sample (Currie et al., 2002).

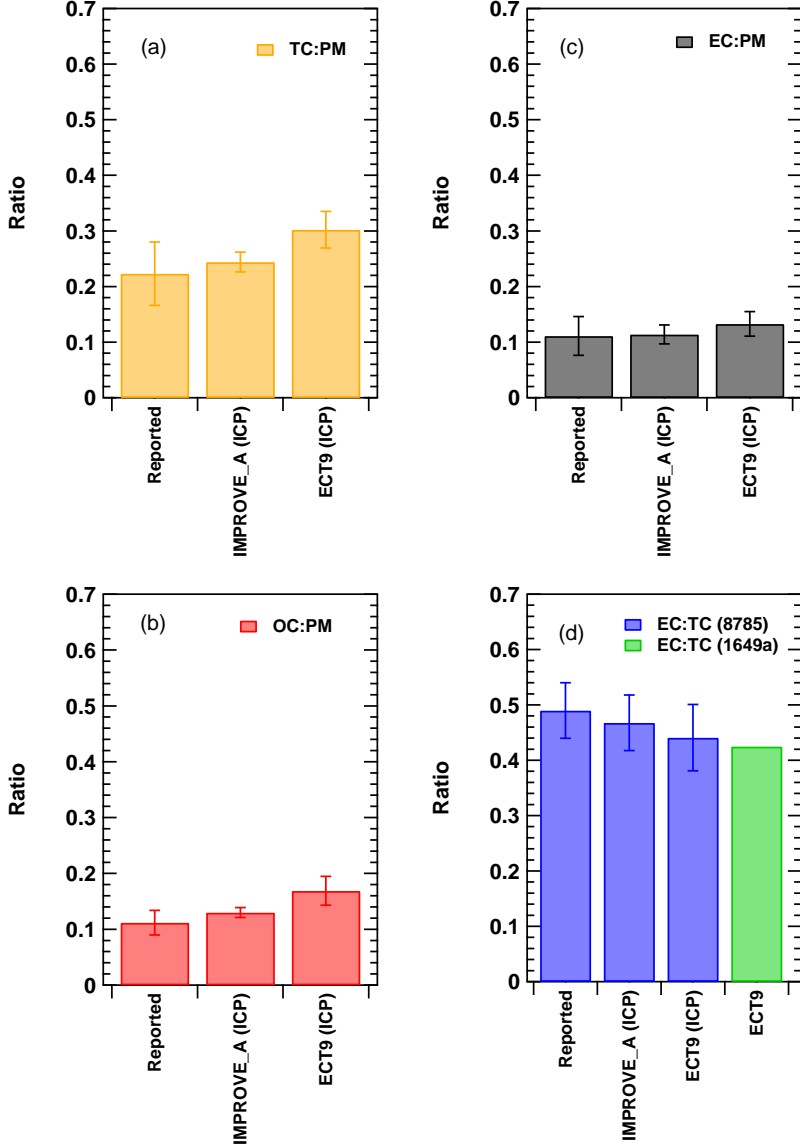






**Figure 3** (a) Real-time Particle Soot Absorption Photometer (PSAP) measurements averaged to match the corresponding sampling frequencies used in different networks. (b) Monthly PSAP measurements derived from (a). (c) Comparison of the different sets of measurements from (b) with the 1:1 line shown in red.

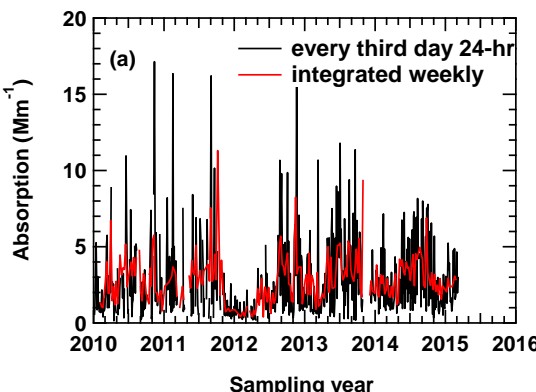

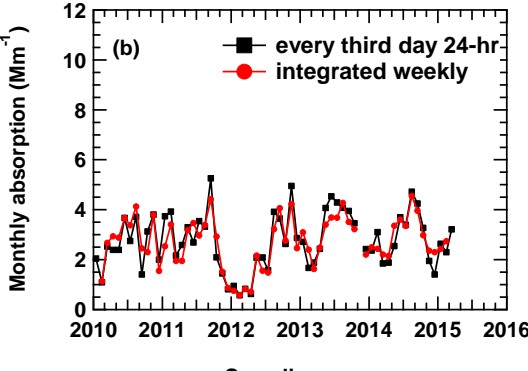

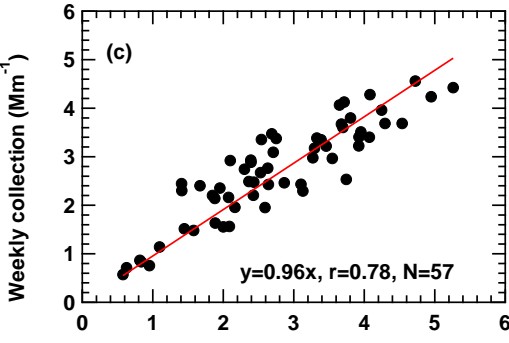



**Figure 4** Monthly averaged CAPMoN (a) OC, (b) EC, and (c) POC mass concentration time series with and
without vapor adsorption correction.  Note that the y-axes in Figures 4b and 4c are on different scale.

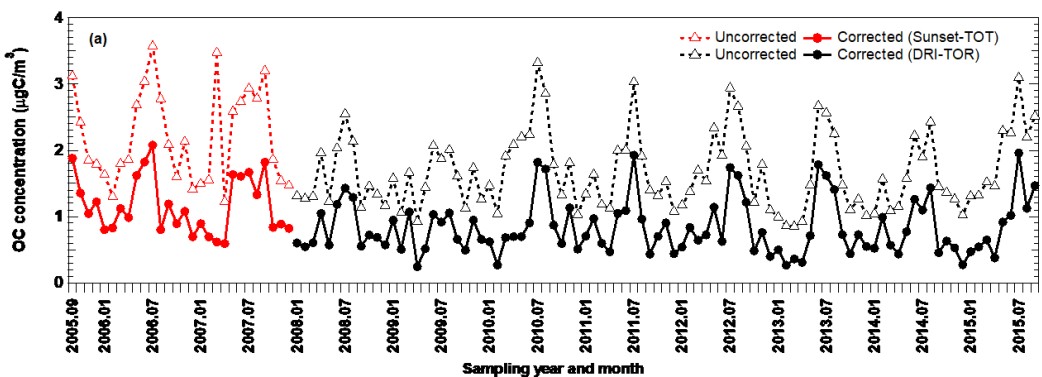


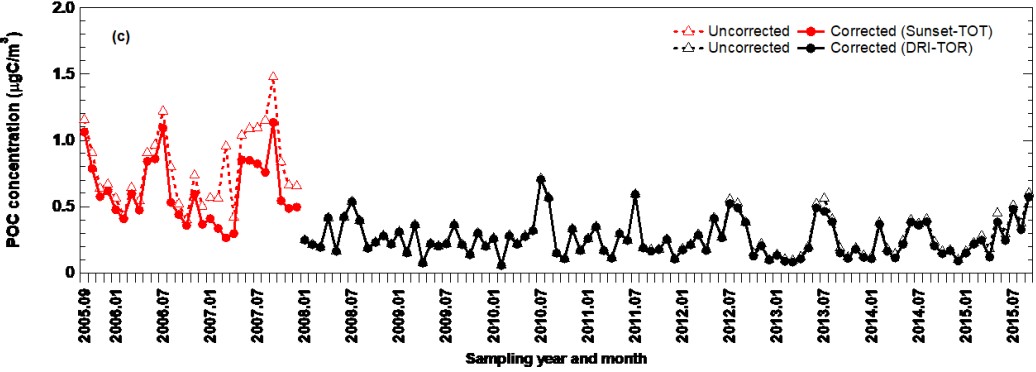









**Figure 5** Relationship between the monthly averaged CAPMoN vapor adsorption corrected and uncorrected measurements for (a) TC, (b) OC, (c) EC, and (d) POC. Black solid markers represent the TOR measurements (2008-2015) analyzed by the DRI analyzer (i.e., DRI-TOR). Red open markers represent the TOT measurements before 2008 analyzed by the Sunset analyzer (i.e., Sunset-TOT). The red line represents the best-fitted linear regression of all the DRI-TOR measurements through the origin. All the corresponding statistics (i.e., best-fitted slope, correlation coefficient, total number of measurement points) are included in the legend.

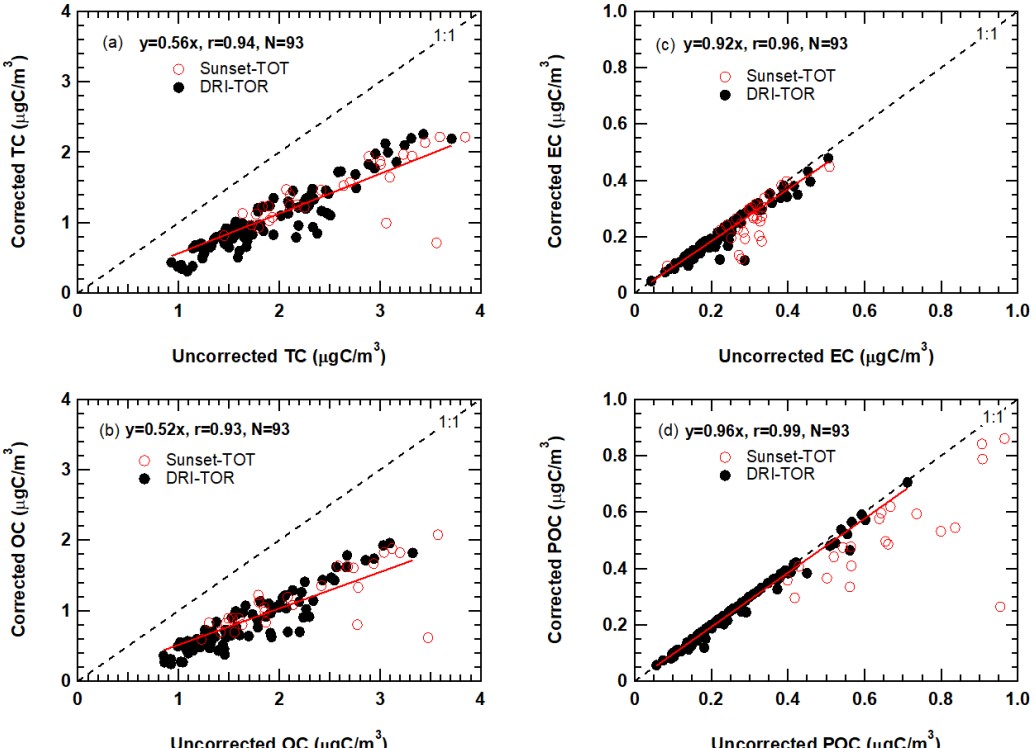



**Figure 6** Monthly averaged (a) TC, (b) OC, (c) EC, and (d) POC concentration time series obtained from three different networks. CAPMoN measurements before 2008 were obtained using Sunset-TOT method (in green) while measurements starting 2008 were obtained using DRI-TOR method (in orange).



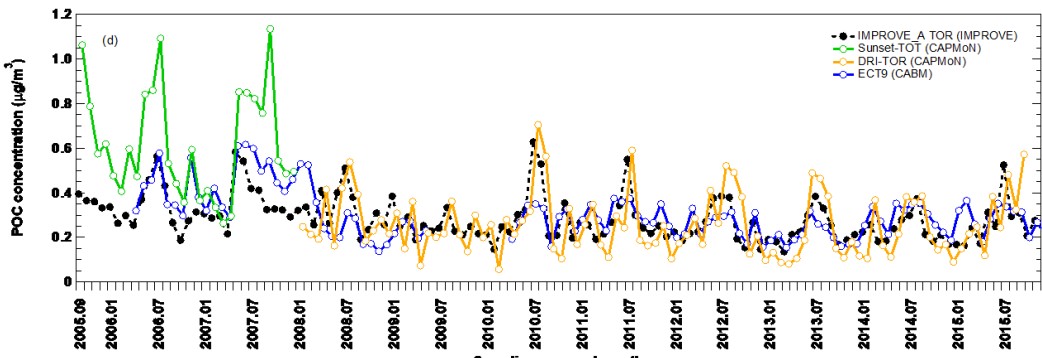








**Figure 7** Comparison of the monthly averaged carbonaceous mass concentrations from the CAPMoN
(red circles and orange triangles) and CABM (black squares) networks against IMPROVE.  The different
straight lines represent the linear regression best fitted line through the origin (i.e., Regression 1).  The
fitted parameters for all corresponding data sets with (Regression 2) and without (Regression 1) the
y-intercept are summarized in Table 2.

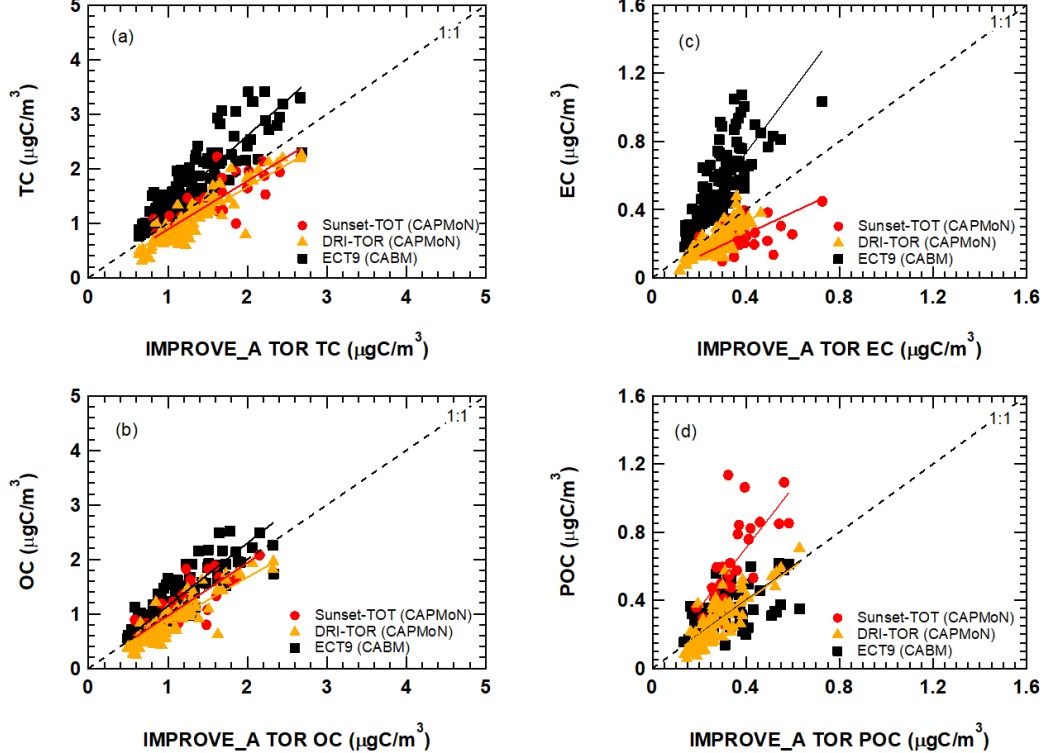


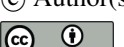



**Figure 8** Normalized monthly averaged time series for (a) TC, (b) OC, (c) EC, and (d) POC for the
IMPROVE (black circles), CAPMoN (orange triangles), and CABM (blue squares) networks. All
measurements were normalized to their corresponding 2008 January mass concentration. Monthly
averaged ambient temperature in green dotted curve. Monthly averaged wind direction and speed (in
wind barb) are in red.

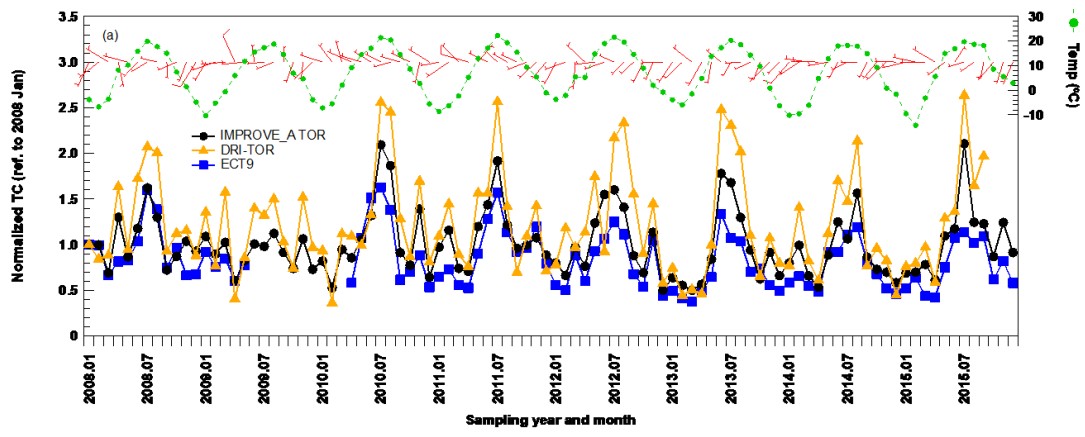


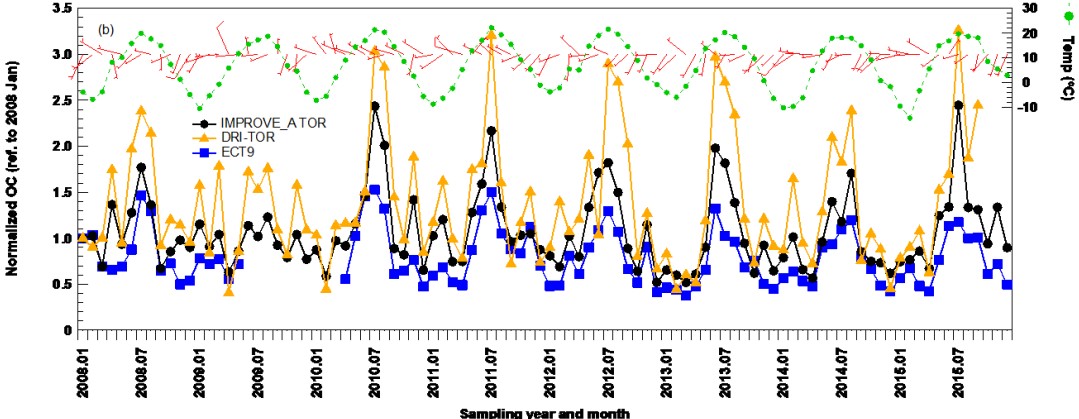


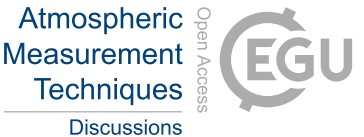



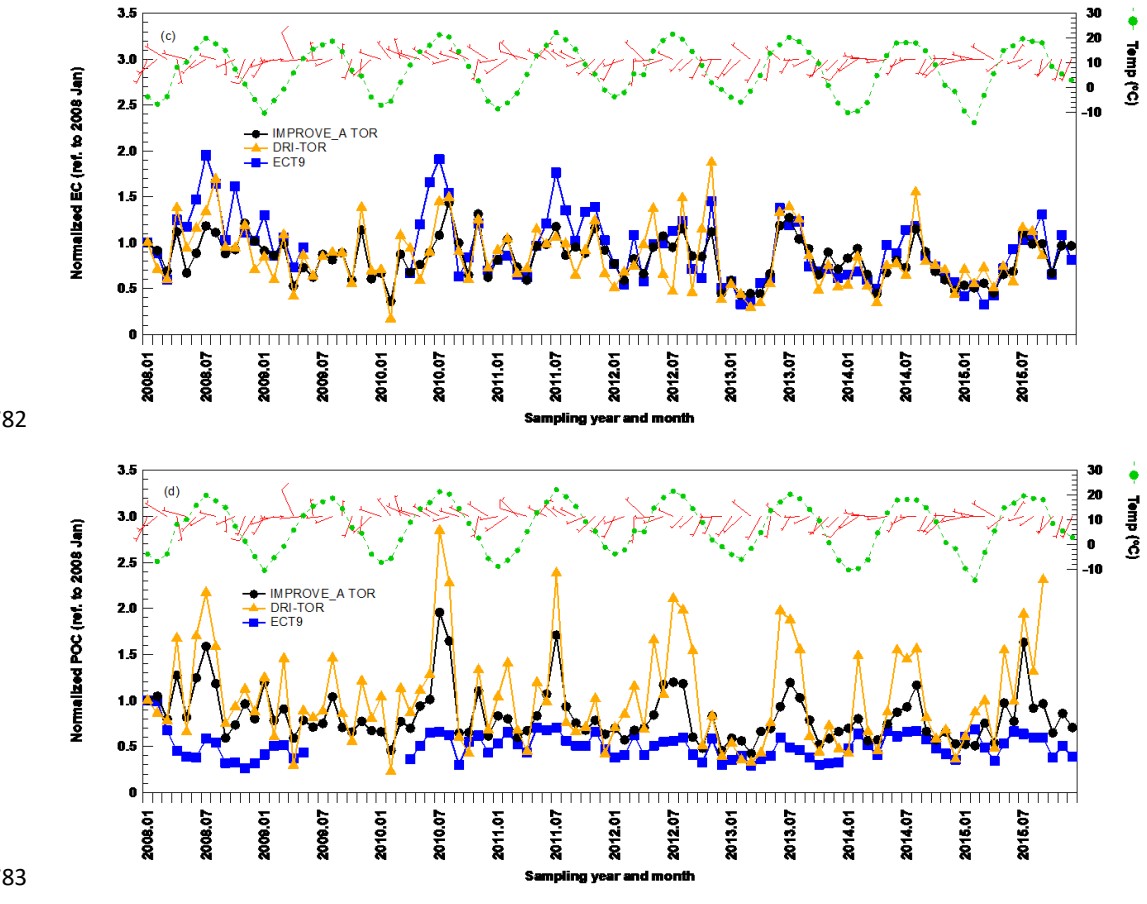









**Figure 9** Comparison of the normalized (a) TC, (b) OC, (c) EC, and (d) POC for the CAPMoN (in circles) and
CABM (in squares) networks against IMPROVE during summer (May-Oct; in yellow) and winter (Nov-Apr;
in black) seasons.  The black solid line represent the best-fitted regression line through the origin of all
measurements.







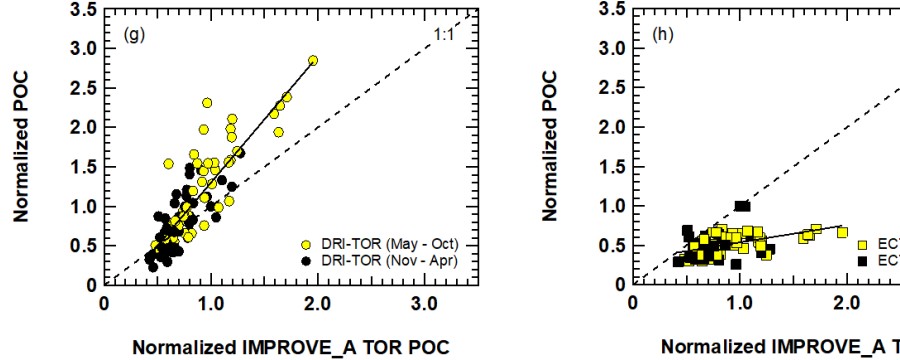



**Figure 10** Figure shows the relationship of averaged (a) TC, (b) OC, and (c) EC concentrations from all
networks as a function of ambient temperature.  Each data point represent the average value of all
network measurements within a 3°C temperature range.  Uncertainties are standard deviations of the
measurements.  Red curve represents the best-fitted Sigmoid function.  Figure 10(d) shows the
seasonality of ECT9 POC compared to the average OC and EC seasonality.  Black solid curve represents
the best-fitted Sigmoid function on all ECT9 POC measurements.

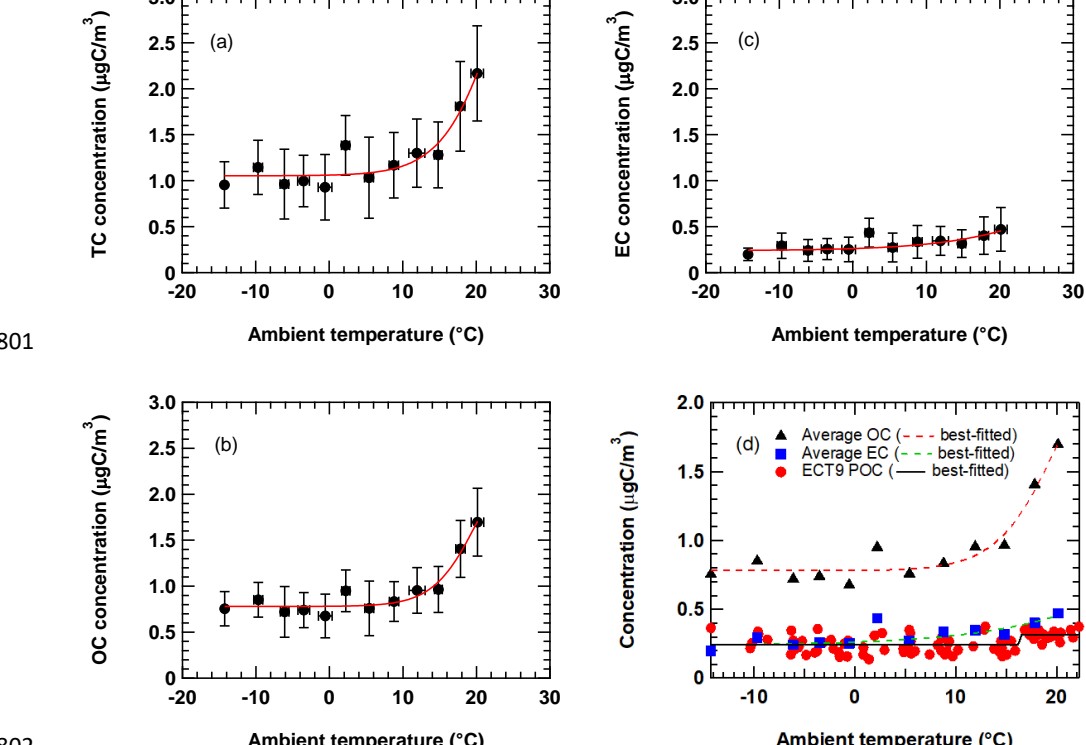

