# Peer review of "Inter-comparison of Elemental and Organic Carbon"

_Atmospheric Measurement Techniques, 2019_

## Referee Comment (RC1) · Anonymous Referee #2 · 14 May 2019

Comments:

This paper describes organic and elemental carbon concentration measured at one site using different sampling devices and flows and using different temperature protocols for analyzing TC, OC and EC. It tries to evaluate how well results are comparable, which is important as it is known that at least the used temperature protocol and used optical correction method have affect to the OC and EC concentrations. Also, different ways of correcting/uncorrecting gaseous artefact were studies.

The paper has clear structure and objective and it is worth of publishing after revision. Detailed comments are described below. In addition to these comments, clarify much more clearly, which results are new and not presented before. Occasionally, it was not clear whose results were presented. Also, check that same tense are mainly used when presented your results. Consider to retitle the subtitles in Results and discussion chapter, some of them were not informative if the names of the network are known.

**Abstract:**

Based on the suggested correction, modify the abstract.

Lines 23-30: OK

Lines 30-32: Why not compared without normalizing the concentrations? Check comment further below. Otherwise this kind of information belongs here.

Lines 32-36: this is not the objective of this paper and not actually studied here. The discussion of the sources of OC and EC is presented in lines 417-425 and are based on other studies. No proves for forest fires occurrence were presented although speculated. Anyway this is not the scope of this study, if I understood correctly. Remove.

Lines 38-41: these lines are more like a conclusions not belong into the abstract.

**Introduction**

Line 43. Modify the sentence to remove double parentheses e.g…carbonaceous aerosol, including elemental carbon (EC), often referred to black carbon (BC) and organic carbon (OC) make up a large fraction…

Line 43-44: reference needed

Lines 120-127, Objective:

- be more specific of how many sites are compared in this study (in line 120-121). Now I got the feeling that multiple sites were compared (line 125)
- remove the names of the networks or write them open

**Sampling and Measurements**

General comments: overall it is slightly difficult to remember the name of the different networks and the used protocols. I need to check them constantly. In the Results and discussion, use other subtitles than the name of the network vs other network.

Line 151-153: Modify the sentence by replacing "The IMPROVE measurements… to Results of/from the IMPROVE measurements

Lines 140-155: Add information if they are sampled at the same day as at the IMPROVE network and add sample amount into Table 1.

Line 155. I do not understand the reference of IMPROVE. Is it a book, paper or internet page? Specify.

Line 166: Re-locate the manufacture info of the quartz filters directly after quartz filters were mentioned.

Lines 163-170: Add information of the sample amount for Sunset-TOT and DRI-TOR and both into table 1.

Line 171-172 Use reference not internet pages for Sunset instrument. Add also, information of the manufactory, and country.

Line 176-178. This sentence is slightly confusing. Are you referring to results presented in Chow et al paper? Modify this sentence more clearly. Inform also what "small difference" means e.g. how much TC mass differs between IMPROVE_A TOR and Sunset-TOT/ DRI-TOR.

Line 179: the subtitle "the ECCC Canadian Aerosol Baseline Network" is slightly confusing as you used name of the CABM network later. Replace to CABM.

Lines 189-191. Add the amount of the filters.

**Differences in Sampling and Analysis among Networks**

Line 213: Modify the sentence so that network is added e.g. cyclones were used in IMPROVE and CABM networks whereas an impactor was used in CAPMoN network.

Line 215: bounce or bounce off. Check, which is correct?

Line 225: Re-order the list so that IMPROVE is before CAPMoN, as it was first introduced in the manuscript.

Lines 225-228: Specify how CAPMoN results (TC, OC, EC) are calculated especially when monthly mean values are presented. Did you use monthly mean value for vapor artifact or did you subtract vapor artifact for individual sample and then calculate the average.

Line 229: References needed after the statement "multiple studies"

**NIST urban dust standard comparison (SRM 8785 & 1649a)**

Remove the NIST and (SRM 8785 and 1649a) from the subtitle

This chapter need to be reorganized and clarified. I did not understand if the intercomparison is the same as the analysis of four replicates.

Reorganize:

Paragraph 1:

Start with the introduction of the urban dust sample (SRM 1649a) then describe how SRM 8785 is done and continue with the reference. After those, describe the intercomparison/analysis of four replicates.

Line 237: "OCEC measurements" is not right way to describe OC and EC analysis. Modify the sentence e.g. consistency between the ECT9 and the IMPROVE_A TOR analytical methods were assessed by measuring four replicates of …….

Line 239: replace IMPROVE_A to IMPROVE_A TOR

Line 240: replace "measuring" to analysing

Line 246-247: ECCC and DRI laboratory has not been presented. Could this information be added under the network presentation e.g. in line 151. Once sampled, filters were stored in freezer until they were ready to be analysed in the DRI laboratory in xx. Similarly the ECCC laboratory.

Paragraph 2: Show first the results based on Figure 2, where analyzed results were compared to the reported one. Change then the numbering of the figures, if Fig. 2 is presented before Fig. 1.

Paragraph 3: Compare TC, OC and EC results analyzed with ECT9 and IMPROVE_A TOR protocols. Were there any test solution that were analyzed during the intercomparison that could indicate the reason of discrepancy (instrumental, inhomogeneous sample etc) of TC between two different protocols and instruments?

Specify whether linear or orthogonal regressions were used in Fig 1. Orthogonal is better if either of the instrument is reference one (and concentrations are known).

Line 251: ….Use correct protocol name "IMPROVE_A TOR" and remove by DRI

Line 251: "compared well" does not inform if the concentration is the same. Modify the sentence.

Paragraph 4

Line 257: clarify what multiple SRM 1649a samples mean. Was it three samples as mentioned in line 266?

Line 267-269: EC to TC ratio of 0.425 measured with carbon isotopes should also compared to the value analyzed with ECT9 protocol. Now it has been compared only for reference value and result derived from the IMPROVE_A TOR protocol.

Line 261: refer that the method is presented in the Supplement material section. You can also consider to present the calculation (Eq 1) and text describing it in the Supplement material section.

**Results and discussion**

Add one paragraph where you have presented how you have compared different samples having different sampling times. If you compare weekly samples to 24h-samples collected every third day, have you calculated average of 2-3 samples and how you have weighted the sampling times to match to the weekly samples as well as possible or have you only compared monthly values. Also, inform if exactly same days

were sampled for IMPROVE and CAPMoN networks. Remind readers that Aug 16, 2006 – Oct 24, 2008 24h-sampling had different sampling times in IMPROVE network than after that.

After this, you can continue with PSAP measurement, but maybe without any subtitle, which is confusing as you have compared PSAP results here. If subtitle is needed, maybe something about "comparability"

PSAP measurement need to be explain under the Sampling and Measurements chapter.

Lines 272-276: Are these results and interpretation presented by Yang et al. or are they interpreted by the writers? Clarify.

Line 279-280: How have the correlation plot in Figure 3c done where weekly and every third day samples were compared? Are the third day samples averaged over 2-3 samples to cover the week samples or are they monthly averages? Clarify.

**Vapor adsorption corrections**

Line 284-286: Why monthly averaged results were presented and not daily? If I understood correctly, artifact correction was made for daily samples. I do understand that it is difficult to present data over long-time period, but clarify how the monthly averages have been calculated. Were artifact subtraction made individually for each sample, which were then averaged over month or calculated first monthly averages of OC and monthly average of gaseous OC and then subtracted. Specify here or in the beginning of the "Results and discussion" chapter.

Lines 285-286: Throughout the paper POC is discussed separately, although it is already included to OC. It is slightly confusing. If not presented/published before, I recommend that one section/paragraph is added where the contribution of POC (monthly averages) from TC for all protocols are presented and discussed.  In addition, POC comparison between 24h TOT (Sunset-TOT) and TOR samples (IMPROVE_A TOR) and between DRI-TOR and IMPROVE_A TOR samples should be done. POC discussion, plots and statistics can be removed elsewhere in the Results and Discussion chapter.

Line 293-294: This sentence is quite loose if the readers have not information of the gaseous artefact of IMPROVE samples. Remove the information presented in lines 309-313 after the information of the CAPMoN samples (line 293). Explain to the readers what anchor IMPROVE sites are (Line 312). It may also be reasonable to remove the blank concentration discussion here after the gaseous artefact discussion.

Add field blank contribution for uncorrected OC values for all three networks. Now, only results of IMPROVE measurements were presented.

Line 296: add detection limit in the unit of ugC/cm2 in parentheses

Line 299: Clarify, why vapor adsorption affects POC correction.

Line 302: remove information in the parentheses (red open circles)

Line 306: add reference after the sentence mentioned of POC to EC ratio. Correct also the mark EC/POC ratio as EC/POC already means a ratio of EC and POC. Discuss of the POC/EC using different protocols and their differences (shortly).

Figure 4: Rescale the y-axis for EC. Remove the POC plot as OC includes the POC.

Figure 5: In this plot, all data points (daily) can be easily presented instead of monthly (, if exactly the same days are sampled). Use daily data and add regression lines and equations for both data sets (DRI-TOR and Sunset-TOT). Use the same color for dots and line for DRI-TOR and another one for Sunset-TOT or color-coded the marks based on the time (or season) for DRI-TOR and Sunset-TOT. Use e.g. gray scale for Sunset-TOT and rainbow scale for DRI-TOR. If too messy, remove one of them to supplement (or make two plots). Also, specify why the linear regression should be go through the zero.

**CAPMoN vs.IMPROVE measurements**

Line 318: Instead of the used subtitle, could it be "comparison of daily sampling methods" or something which describes more illustratively what is compared, if the networks are not familiar for the readers.

Lines 319-321: The discussion of summer peak should be removed to the chapter Seasonality in Carbon.

Lines 321-326: the correlation coefficients have been presented in the table 2, do not repeat the values in the text. Concise these lines e.g. better correlations of TC, EC and OC were found between the protocols that use same POC correction method (DRI-TOR and IMPROVE_A TOR) than between Sunset-TOT, which use transmittance for POC correction and IMPROVE_A TOR (Table 2). Especially correlation of EC between Sunset-TOT and IMPROVE_A TOR was poor. Note, that Sunset-TOT and IMPROVE_A TOR had slightly different sampling time.

Figure 6: CAPMoN time series have been already presented in Fig 4. Remove this figure and plot correlation plot between IMPROVE and CAPMoN 24h-measurements (, if exactly the same days are sampled) instead of monthly mean. Color-code the marks based on season/time/or something else.

Lines 327-331 and Table 2: Clarify what kind of regression (linear, orthogonal) has been used. Prefer orthogonal. Clarify also, are the regression calculated from monthly mean values? Remove slopes, which are already presented in Table2.  Explain why Regression 1 was used. Is it correct to force through the zero?

Lines 332-336: remove this paragraph to the new POC section.

**CABM vs. IMPROVE measurements**

Line 337: Change the subtitle e.g. Monthly comparison or something else

As CABM measurements does not subtract the gaseous artefact, the writers may consider to plot figures 6 and 7 with uncorrected data.

Figure 6: remove 6a-c to Supplement and delete 6d. Modify Fig 6a-c so that common x-axis is used to save space. Refer also to Figure 7 that should be presented against (x-axis) to CABM network that has the different sampling time compared to other networks.

Line 340-341: after "comparable" should be present correlation coefficient. The percentage shows the similarity of the concentrations. Also, if it is said that concentrations are higher, the writers should said where to compare "higher than". Modify this sentence.

Lines 342-346: Again, I do not understand why both regressions are presented. Why fits are forced through the origin? I recommend to use only regression with intercept unless there is a clear reason for forcing through the zero. Again if comparative is used, there have to be the other party.

Lines 351-352: CABM network did not see any short-term variation as it has week-long sampling time. Anyway the Fig. S3 shows monthly mean values that is even longer time than week. Modify the sentence.

Line 353-359: POC discussion should be remove to its own section/paragraph. In line 356, Table 3 has not been presented yet. Why not use table 2? Remove the regression discussion with forced intercept

**Comparison of the Normalized Time Series**

I do not understand this chapter. Why the data should be normalized to Jan 2008 data? This can be removed or explained better the meaning of this chapter.

**Seasonality in Carbon**

Although this chapter if very interesting, it is not part of the objective. To stay with the objective, it would be interesting to see how the gaseous artefact correction varies between the season/temperature for DRI-TOR, Sunset-TOT and IMPROVE_A TOR. The writers can use the Sigmoid function if wanted but concentrate on the contribution of gaseous artefact. Also comparison of TC, OC and EC during different season between different networks is interesting. Is there any differences between different networks based on the season?

In the beginning of the Results and Discussion, the writers can present general overview of the results (yearly concentrations of TC, OC and EC). In addition, wind roses and footprints, if wanted, can be presented shortly here and put the plots in the Supplement.

**Conclusions:**

Based on the modification and comments, modify this chapter. In addition

Line 443-444: the filter face velocity does not affect for the field blank concentration. Now the readers may get wrong message. Modify. Add also information of the other field blanks e.g. field blanks accounted xx-xx% (xx-xx ugC/cm2) of the measured OC.

Lines 445-446: I am not sure if this statement was proved in this manuscript, although true. Modify this sentence e.g. Start with the information that CABM network did not correct gaseous artefact and its OC has xx% higher concentration than with two other networks that had the correction done.

Lines 446-448: I do not understand this statement or its purpose. Too long story and too much details (like values of r). Describe the contribution of POC of TC /OC.

Lines 451-452: SRM 8785 samples demonstrate the consistency of the different protocols not long-term carbon measurements. Correct.

Line 457: "North American harmonized carbonaceous concentration map" is new for me and may be to other readers. Clarify shortly

**Supplement:**

Line 69-70. Actually internal signal is used to correct slight variation during each analysis. TC, OC and EC concentrations are calculated based on the calibration value calculated from external calibration. Correct.

Line 79: There were no IMPROVE protocol. It was named to DRI-TOR and Sunset-TOT. Correct

Table S1: Use the protocols names you have chosen to use (Sunset-TOT and DRI-TOR). First column IMPROVE_A TOR, second column both Sunset-TOT and DRI-TOR and third column ECT9

FigureS1: protocol name IMPROVE has been used, although not mentioned in the manuscript. Either rename that or add to the manuscript that the temperature steps used in analyzing particulate carbon in CAPMoN network are called IMPROVE although different optical correction used.

Table S1 and Figure S1: Replace IMPROVE_A to IMPROVE_A TOR

Figure S3: Add information that results are monthly averages.

---

## Referee Comment (RC2) · Anonymous Referee #1 · 16 May 2019

Review of "Inter-comparison of the elemental and organic carbon mass measurements from three North American national long-term monitoring networks" by Chan et al.

This paper summarizes collocated organic and elemental concentrations from three different types of analysis and sampling protocol. The results are useful in furthering our understanding of thermal optical analyses and resulting biases from sampling artifacts as well as temperature protocols. The paper is fairly well organized and written but could benefit from some clarification. I recommend publication after the authors address comments below.

[Figure]

Line 1: The title, as well as some description in the text is somewhat misleading because it implies that large geographic scale comparisons are made when in fact the comparisons only exist at one site. Perhaps changing or including something regarding different analytical protocols would help clarify this point.

Line 23: Please state years.

Line 23-26: Again, similar to the title, point out that collocated samplers only exist at one site, so really what is being compared here are the impacts from different sampling and analytical protocols, not a large scale geographic comparison.

Line 29: More on this later, but I don't understand the value of the normalized comparison. The agreement depends on what you have normalized each time series to. Over what time periods where these comparisons made?

Line 35: Is there any evidence for linkages to forest fire emissions and increased vehicular emissions? Did the authors include analysis of these emissions or is this conjecture?

Line 36-37: This may be true depending on artifact corrections and how they are applied across a network.

Line 38: Again, extrapolating data and comparisons from one site to "regional to continental-scale-harmonized maps" hasn't been shown here and may not necessarily be true, especially given different sampling times and sources.

Line 23-41: I think it would be helpful if the abstract more closely reflected the comparison work rather than sources which was a rather minor part of the work and mostly based on previously published work (e.g., secondary aerosol formation in summer, smoke in summer, higher OC and EC in summer, etc.).

Line 52: OC can also absorb solar radiation.

Line 52,53,56,57: I would suggest using either BC or EC and keeping the same nomenclature throughout the paper, unless the authors are actually referring to different measurements, then clarify and define.

Line 58: Include "impacts of" changing emissions since OC and EC measurements don't directly determine emissions.

Line 61: The first sentence is unclear. Wouldn't long term measurements just depend on making the same measurement over time and doesn't really depend on a universal definition?

Line 67: The sentence starting with "BC is a generic term" would be a better starting sentence for this paragraph and the authors could remove the current first sentence.

Line 71: Replace "being" with "is"

Line 72: Include "as" after "EC is referred to"

Line 75: Can the authors clarify what they mean by "EC and BC resembled each other"?

Line 80: I am not sure what the authors mean by "direct measurement of carbon mass as part of gravimetric mass"?

Line 97: I'm not sure what is meant by "resulting EC method"?

Line 111: Can the authors provide a reference for the OC overestimation?

Line 120: The acronyms for the various networks should be spelled out at first usage.

Line 122: Again, this is somewhat misleading. Add that these collocated measurements occurred at one site.

Line 123: I might have missed this later, but what are the solutions for improving the compatibility?

Line 124: I am not sure the results from one site have been demonstrated to create a regional and continental scale harmonized carbon concentration data set.

Line 138: What is meant by "regional-scale monitors"? Do the authors mean that many samplers operate across the United States?

Line 139: replace "understanding long-range transport" with "understanding long-term trends".

Line 139-140: I suggest replacing the Malm 1989 reference with the Malm 1994 reference (Malm, W. C., J. F. Sisler, D. Huffman, R. A. Eldred, and T. A. Cahill (1994), Spatial and seasonal trends in particle concentration and optical extinction in the United States, J. Geophys. Res., 99(D1), 1347-1370)

Line 143-144: The IMPROVE samplers typically sample midnight to midnight, was the sampler at Egbert running on a different schedule?

Line 148: Are the filters shipped cold?

Line 155: Spell out CAPMoN.

Line 159: Do the measurements include carbon at all of these sites as well?

Line 176: Also see Malm et al. (2001) for a discussion of sampling biases on OC and EC concentrations (Malm, W. C., B. A. Schichtel, and M. L. Pitchford (2011), Uncertainties in PM2.5 gravimetric and speciation measurements and what we can learn from them, J. Air & Waste Manage. Assoc., 61, 1131-1149, doi:10.1080/10473289.2011.603998.)

Line 179: Spell ECCC- Also, please choose notation, either CABM or ECCC. Both are used interchangeably throughout the paper and it is confusing.

Line 184: replace "costal" with "coastal"

Line 207: replace "measurements is" with "measurements are"

Line 211: Include "an" between "uses" and "impactor"

Line 212: Replace "Impactor" with "Impactors"

Line 218-222: See the Malm et al., 2011 paper mentioned earlier.

Line 228: Can the authors provide some references for the multiple studies?

Line 232: Change "introduce" to "introduces"

Line 236: I think you can remove "SRM 8785 & 1649a" from the section header.

Line 246: No correlations are given in Figure 1. Also include figure parts in the text and include OC.

Line 248-9: Need figure parts for Figure 2 in the text too (e.g., Figure 2(a)-(d) shows TC, EC, OC, and EC/TC, respectively)

Line 249-250: It is unclear what the authors mean by "Irrespective of data disparity"?

Line 270: I am not sure this is clear: Do the authors just resample the high resolution data for different averaging times? When they say different data sets do they mean the same measurement just with different averaging times? Wouldn't you expect these to compare well? Or do they compare EC to the PSAP measurement? The figures have units of Mm-1, so it suggests that they either converted EC to absorption coefficient (if so, what absorption efficiency was used?). Please clarify, including figure caption 3 when "comparison of different sets of measurements" from (c) because it is misleading.

Line 255: Which EC/TC value was further verified? Also, replace "sample" with "samples"

Line 283: Over what time period?

Line 288: Can the authors comment on the offset (nonzero intercept) and what it implies in terms of sampling artifacts or biases?

Line 298: Yes, the POC correction directly influences EC concentrations. Can the authors comment on this vapor adsorption issue with respect to the PSAP weekly comparisons?

Line 303: Add a period and start "An optical correction" as a new sentence.

Line 317: Include "monthly mean" before DRI-TOR CAPMon measurements and "comparable to the concentrations derived from the IMPROVE_A. . .."

Line 320: What are considered "good correlations"?

Line 351-352: Can the authors describe Figures 7a-c before 7d to keep them in order?

Line 355: At what level of significance?

Line 358: I am not convinced the normalized analysis is necessary and adds to the paper. The comparisons between samplers would change depending on what the data are normalized to (choose a different month or an annual mean for example). The comparisons already discussed are more useful because they show the true biases. The diurnal wind cycles on the timelines could be added to the earlier timelines if the authors want to include that analysis.

Line 393: I think elevated carbon concentrations in summer are better shown in Figure 6 given the averaging times.

Line 413: When are the concentration in the N and NW higher?

Line 414: Do the authors mean residential instead of residual?

Line 431: How appropriate is the comparison with ECT9 POC since this is a nonlinear relationship?

Line 447: Also include longer sampling time.

Line 452: What are typical measurement uncertainties? Are these greater?

Line 452: Note that others have performed similar comparisons across networks (CSN and IMPROVE) for continental scale integration. Biases for both OC and EC between networks were less than 10% (similar sampling and analytical procedures). Hand, J. L., B. A. Schichtel, M. Pitchford, W. C. Malm, and N. H. Frank (2012a), Seasonal composition of remote and urban fine particulate matter in the United States, J. Geophys. Res., 117, D05209, doi:10.1029/2011JD017122. Hand, J. L., B. A. Schichtel, W. C. Malm, and N. H. Frank (2013), Spatial and temporal trends in PM2.5 organic and elemental carbon across the United States, Advances in Meteor., 2013, Article ID 367674.

Line 504: This link did not work, it needs to be updated: http://vista.cira.colostate.edu/Improve/improve-data/

Line 699: Table 1. Can the authors clarify: Is "IMPROVE" under CAPMoN consistent with lines 170-171 that lists IMPROVE-TOT for 2005-2007 and IMPROVE_TOR protocol? It is challenging to keep these different protocols straight and so careful attention to how they are referred to in the paper and the tables would help.

Line 702: Table 2: Similar comment, here it is referred to as "Sunset-TOT". The number of significant digits included in this table seem unnecessary.

Line 710, 713: I don't think these tables are necessary, see earlier comment.

Line 718: Figure 1: Again, please be consistent with ECCC and CABM

Line 725: Figure 2, Same comment as previous figure. What is ICP? Please relate x-axis labels to the caption description.

Line 732: Figure 3: See earlier comment- this comparisons is unclear.

Line 740: Figure 4: It would help to see the comparisons in (b) and (c) if the scales were reduced. Again, note the data description in the figures do not match the discussions or tables (e.g., "Sunset-TOT")

Line 757: Figure 6: Please include location in this figure caption so it is clear that the three different networks are collocated at one site.

Line 766: Figure 7: Which "IMPROVE" are the comparisons made against? Please be clear in the caption to match the axis labels.

Figures 8 and 9: Are unnecessary and do not lend to a better understanding of the comparisons.

---

## Author Comment (AC1) · 18 Jul 2019

Comments:
This paper describes organic and elemental carbon concentration measured at one site using different sampling devices and flows and using different temperature protocols for analyzing TC, OC and EC. It tries to evaluate how well results are comparable, which is important as it is known that at least the used temperature protocol and used optical correction method have affect to the OC and EC concentrations. Also, different ways of correcting/uncorrecting gaseous artefact were studies.
The paper has clear structure and objective and it is worth of publishing after revision. Detailed comments are described below. In addition to these comments, clarify much more clearly, which results are new and not presented before. Occasionally, it was not clear whose results were presented. Also, check that same tense are mainly used when presented your results. Consider to retitle the subtitles in Results and discussion chapter, some of them were not informative if the names of the network are known.
>> The authors appreciate the useful comments and suggestions from the referee, and we address all the comments accordingly.

**Abstract:**
Based on the suggested correction, modify the abstract.
>> This is addressed.

Lines 23-30: OK
Lines 30-32: Why not compared without normalizing the concentrations? Check comment further below. Otherwise this kind of information belongs here.
>> After some consideration and discussions among the co-authors, we agree to remove this section and focus on the absolute data inter-comparison section.

Lines 32-36: this is not the objective of this paper and not actually studied here. The discussion of the sources of OC and EC is presented in lines 417-425 and are based on other studies. No proves for forest fires occurrence were presented although speculated. Anyway this is not the scope of this study, if I understood correctly. Remove.
>> The authors have conducted some preliminary analysis and results suggested that forest fire could potentially influence the Egbert site and result in elevated EC concentration during summer time. Additional research is currently on going and the results are expected to be included in a separate manuscript. Considering this is preliminary results, we have now removed such content to the supporting material.

Lines 38-41: these lines are more like a conclusions not belong into the abstract.
>> These sentences are removed.

**Introduction**
Line 43. Modify the sentence to remove double parentheses e.g…carbonaceous aerosol, including elemental carbon (EC), often referred to black carbon (BC) and organic carbon (OC) make up a large fraction…
>> This is addressed.

Line 43-44: reference needed
>> A reference has been added.

Lines 120-127, Objective:

• be more specific of how many sites are compared in this study (in line 120-121). Now I got the feeling that multiple sites were compared (line 125)
• remove the names of the networks or write them open
>> This is addressed. We have made it clear that the comparison was not for multiple sites.

**Sampling and Measurements**
General comments: overall it is slightly difficult to remember the name of the different networks and the used protocols. I need to check them constantly. In the Results and discussion, use other subtitles than the name of the network vs other network.
>> The authors apologize for the confusion. The protocol names throughout the paper have now been verified and modified to ensure they are consistent. The subtitles have also been revised to avoid confusion.

Line 151-153: Modify the sentence by replacing "The IMPROVE measurements… to Results of/from the IMPROVE measurements
>> We have improved the sentence.

Lines 140-155: Add information if they are sampled at the same day as at the IMPROVE network and add sample amount into Table 1.
>> The CAPMoN samples were indeed collected on the same day as the IMPROVE samples and this information is now mentioned in the manuscript. We have also included the total number of samples used in the analysis in Table 1.

Line 155. I do not understand the reference of IMPROVE. Is it a book, paper or internet page? Specify.
>> We have removed this reference.

Line 166: Re-locate the manufacture info of the quartz filters directly after quartz filters were mentioned.
>> This is addressed.

Lines 163-170: Add information of the sample amount for Sunset-TOT and DRI-TOR and both into table 1.
>> The number of samples are now added.

Line 171-172 Use reference not internet pages for Sunset instrument. Add also, information of the manufactory, and country.
>> Manufactory is Sunset Laboratory Inc. from the USA. This info is now added.

Line 176-178. This sentence is slightly confusing. Are you referring to results presented in Chow et al paper? Modify this sentence more clearly. Inform also what "small difference" means e.g. how much TC mass differs between IMPROVE_A TOR and Sunset-TOT/ DRI-TOR.
>> Yes, we are referring to the results discussed in Chow et al. (2007) and the sentence is revised for clarity. The "small difference" refers to the temperature difference discussed in the previous sentence, which is the typical temperature difference between each ramping temperature used in the two protocol.

Line 179: the subtitle "the ECCC Canadian Aerosol Baseline Network" is slightly confusing as you used name of the CABM network later. Replace to CABM.

>> This is addressed.

Lines 189-191. Add the amount of the filters.
**>>** The total number of samples has been included in Table 1.

**Differences in Sampling and Analysis among Networks**
Line 213: Modify the sentence so that network is added e.g. cyclones were used in IMPROVE and CABM networks whereas an impactor was used in CAPMoN network.
>> This information is included.

Line 215: bounce or bounce off. Check, which is correct?
>> We mean bounce off. When hit the impactor surface, some large solid particles may bounce and not be collected by the impactor plate and then re-enter the airstream and be collected by the filter downstream.

Line 225: Re-order the list so that IMPROVE is before CAPMoN, as it was first introduced in the manuscript.
>> This is addressed.

Lines 225-228: Specify how CAPMoN results (TC, OC, EC) are calculated especially when monthly mean values are presented. Did you use monthly mean value for vapor artifact or did you subtract vapor artifact for individual sample and then calculate the average.
>> For CAPMoN measurements, vapor adsorption artifact was applied to each individual 24-h samples. Then, all the artifact corrected samples within each month were used to compute the monthly average measurement. For IMPROVE measurements, the monthly median OC artifact derived from 13 sites were subtracted from all individual OC measurements in the same month before monthly averaged were derived. The above information is now added to the manuscript.

Line 229: References needed after the statement "multiple studies"
**>>** The "multiple studies" here were indeed referring to the references in line 231 (i.e., Chow et al., 2004; 2005; Watson et al., 2005). The sentence is now revised to include references.

**NIST urban dust standard comparison (SRM 8785 & 1649a)**
Remove the NIST and (SRM 8785 and 1649a) from the subtitle
>> This is addressed.

This chapter need to be reorganized and clarified. I did not understand if the intercomparison is the same as the analysis of four replicates.
>> The word "inter-comparison" was used because this is a comparison exercise conducted by two labs even though there were just four replicates. It was realized that using "replicates" was not proper in description of those SRM 8785 filters since they are not the same in mass loading. This has been addressed in the revised version.

Reorganize:
Paragraph 1:
Start with the introduction of the urban dust sample (SRM 1649a) then describe how SRM 8785 is done and continue with the reference. After those, describe the intercomparison/analysis of four replicates.

Line 237: "OCEC measurements" is not right way to describe OC and EC analysis. Modify the sentence e.g. consistency between the ECT9 and the IMPROVE_A TOR analytical methods were assessed by measuring four replicates of …….
>> This is addressed and the paragraph is now rearranged.

Line 239: replace IMPROVE_A to IMPROVE_A TOR
>> This is addressed.

Line 240: replace "measuring" to analysing
>> This is addressed.

Line 246-247: ECCC and DRI laboratory has not been presented. Could this information be added under the network presentation e.g. in line 151. Once sampled, filters were stored in freezer until they were ready to be analysed in the DRI laboratory in xx. Similarly the ECCC laboratory.
>> This is addressed.

Paragraph 2: Show first the results based on Figure 2, where analyzed results were compared to the reported one. Change then the numbering of the figures, if Fig. 2 is presented before Fig. 1.
>> This is addressed and Fig 1 and 2 are now in reversed order.

Paragraph 3: Compare TC, OC and EC results analyzed with ECT9 and IMPROVE_A TOR protocols. Were there any test solution that were analyzed during the intercomparison that could indicate the reason of discrepancy (instrumental, inhomogeneous sample etc) of TC between two different protocols and instruments?
>> Unfortunately, no such solution was analyzed by both labs in this inter-comparison effort. The current analysis was not able to determine the reason for causing the difference observed during the inter-comparison. During the analysis, both labs analyzed the filters using their own standard operation procedure and therefore the regression results reflect any difference that would be caused by all reasons in combined.

Specify whether linear or orthogonal regressions were used in Fig 1. Orthogonal is better if either of the instrument is reference one (and concentrations are known).
>> We have specify the type of regressions to use in the revised version.

Line 251: ….Use correct protocol name "IMPROVE_A TOR" and remove by DRI
>> This is corrected.

Line 251: "compared well" does not inform if the concentration is the same. Modify the sentence. Paragraph 4
>> "Compared well" means the average values were within uncertainties and therefore they are not statistically different. The sentence is now modified by stating this explicitly.

Line 257: clarify what multiple SRM 1649a samples mean. Was it three samples as mentioned in line 266?
>> We literally means a few. In here, SRM 1649a (which is dust powder) were weighted and analyzed by the OCEC analyzer for TC, and then separately for OC and EC. The word "multiple" is removed to avoid confusion.

Line 267-269: EC to TC ratio of 0.425 measured with carbon isotopes should also compared to the value analyzed with ECT9 protocol. Now it has been compared only for reference value and result derived from the IMPROVE_A TOR protocol.
>> This is addressed.

Line 261: refer that the method is presented in the Supplement material section. You can also consider to present the calculation (Eq 1) and text describing it in the Supplement material section.
>> We prefer to leave a brief discussion of the method in the main text while all the technical details of the methodology will remain in the supplement material section.

**Results and discussion**
Add one paragraph where you have presented how you have compared different samples having different sampling times. If you compare weekly samples to 24h-samples collected every third day, have you calculated average of 2-3 samples and how you have weighted the sampling times to match to the weekly samples as well as possible or have you only compared monthly values. Also, inform if exactly same days were sampled for IMPROVE and CAPMoN networks. Remind readers that Aug 16, 2006 – Oct 24, 2008 24h-sampling had different sampling times in IMPROVE network than after that.
After this, you can continue with PSAP measurement, but maybe without any subtitle, which is confusing as you have compared PSAP results here. If subtitle is needed, maybe something about "comparability"
>> The corresponding paragraph is revised to include more information regarding how the comparison is done.

PSAP measurement need to be explain under the Sampling and Measurements chapter.
>> This is addressed.

Lines 272-276: Are these results and interpretation presented by Yang et al. or are they interpreted by the writers? Clarify.
>> The results (comparison between the integrated weekly and once every third day samples) are conclusions from Yang et al. This is clarified in the revised version.

Line 279-280: How have the correlation plot in Figure 3c done where weekly and every third day samples were compared? Are the third day samples averaged over 2-3 samples to cover the week samples or are they monthly averages? Clarify.
>> Results in Fig 3c represent the comparison between the two sets of monthly averages derived from the integrated weekly and once every third day samples. This is clarified in the revised version.

**Vapor adsorption corrections**
Line 284-286: Why monthly averaged results were presented and not daily? If I understood correctly, artifact correction was made for daily samples. I do understand that it is difficult to present data over long-time period, but clarify how the monthly averages have been calculated. Were artifact subtraction made individually for each sample, which were then averaged over month or calculated first monthly averages of OC and monthly average of gaseous OC and then subtracted. Specify here or in the beginning of the "Results and discussion" chapter.
>> For the CAPMoN measurements, artifact correction was applied to the 24-hour samples. Then the artifact corrected data were averaged over the month to get the monthly average. For downloaded IMPROVE measurements were already artifact corrected. Average vapor adsorption in a monthly basis

was first determined from measurements from 13 sites (exclude Egbert). Then, such value was applied to all individual measurements before the monthly average is computed.

Monthly data is used here to assess the comparability among three networks. The original measurements from various networks have different sampling frequencies (every three day vs. weekly integrated) and it could cause complications. Thus, monthly averages are used to be consistent through the entire manuscript. In addition, monthly means are often considered as a reliable time resolution in comparisons between climate models and observations, due to the limitation of reported emission inventories (usually as annual values). Therefore, the analysis obtained here could be directly relevant to those comparisons . We have included a statement about how the measurements presented in this section were obtained.

Lines 285-286: Throughout the paper POC is discussed separately, although it is already included to OC. It is slightly confusing. If not presented/published before, I recommend that one section/paragraph is added where the contribution of POC (monthly averages) from TC for all protocols are presented and discussed. In addition, POC comparison between 24h TOT (Sunset-TOT) and TOR samples (IMPROVE_A TOR) and between DRI-TOR and IMPROVE_A TOR samples should be done. POC discussion, plots and statistics can be removed elsewhere in the Results and Discussion chapter.
>> POC from IMPROVE_A TOR and DRI-TOR are simply a charring correction and this analysis also show that it is always proportional to OC. On the other hand, ECT9 POC is not a charring correction but appear to represent a different class of organics, likely the oxygenated organics. For the ECT9 method, POC is considered as a separate carbonaceous fraction from the measured OC although reported as part of "total OC". To a certain extent, the POC from various method were compared through the use of correlation coefficient.

Line 293-294: This sentence is quite loose if the readers have not information of the gaseous artefact of IMPROVE samples. Remove the information presented in lines 309-313 after the information of the CAPMoN samples (line 293). Explain to the readers what anchor IMPROVE sites are (Line 312). It may also be reasonable to remove the blank concentration discussion here after the gaseous artefact discussion.
>> The reason to include the artifact information for the IMPROVE samples is to verify the statement mentioned earlier that the lower filter face velocity of the CAPMoN measurements leads to higher filter artifact. We believe the content here provides readers a perspective of the relative magnitude of the artifact when dealing with the different measurements. Also, IMPROVE has changed their SOP and use blank correction to address the artifact correction for new measurements. Although this does not impact the measurements used in this manuscript, we thought it was a good idea to include such information. This paragraph is revised to avoid confusion. The filter blank concentration discussion is now removed.

Add field blank contribution for uncorrected OC values for all three networks. Now, only results of IMPROVE measurements were presented.
>> We decided to leave out the discussion of filter blank because this is not handled the same across different networks.

Line 296: add detection limit in the unit of ugC/cm2 in parentheses
>> This is addressed.

Line 299: Clarify, why vapor adsorption affects POC correction.

>> As seen in Figure 4 and Figure S2, the backup filter also possesses a small amount of POC and therefore artifact lowers the POC concentration slightly, however, the magnitude of the POC artifact has never come close to the artifact for OC.

Line 302: remove information in the parentheses (red open circles)
>> This is addressed.

Line 306: add reference after the sentence mentioned of POC to EC ratio. Correct also the mark EC/POC ratio as EC/POC already means a ratio of EC and POC. Discuss of the POC/EC using different protocols and their differences (shortly).
>> This is addressed.  The authors did not intend to introduce another parameter (POC/EC).  Although this was used in the reference Chen et al. (2004).  What the authors intended to say here is that an optical correction using reflectance is a more consistent method than the optical correction using transmission under the situation when POC concentration is large compared to EC.  We have revised the content here accordingly to avoid the confusion.

Figure 4: Rescale the y-axis for EC. Remove the POC plot as OC includes the POC.
>> Figure 4b has been rescaled.  Although POC is part of OC (which is now mentioned in the text), the authors would like to retain Figure 4c in the text.  This is to illustrate the point that although artifact influence the POC concentration and therefore impact EC concentration indirectly, the influence is small hand artifact affects only OC primarily.

Figure 5: In this plot, all data points (daily) can be easily presented instead of monthly (, if exactly the same days are sampled). Use daily data and add regression lines and equations for both data sets (DRI-TOR and Sunset-TOT). Use the same color for dots and line for DRI-TOR and another one for Sunset-TOT or color-coded the marks based on the time (or season) for DRI-TOR and Sunset-TOT. Use e.g. gray scale for Sunset-TOT and rainbow scale for DRI-TOR. If too messy, remove one of them to supplement (or make two plots). Also, specify why the linear regression should be go through the zero.
>> We have addressed this in the previous comment.

**CAPMoN vs.IMPROVE measurements**
Line 318: Instead of the used subtitle, could it be "comparison of daily sampling methods" or something which describes more illustratively what is compared, if the networks are not familiar for the readers.
>> The authors agree that this title may not be as appropriate.  We have now combine the section "CAPMoN vs. IMPROVE measurements" and "CABM vs. IMPROVE measurements" to one paragraph titled "Comparison among IMPROVE, CAPMoN, and CABM Measurements".

Lines 319-321: The discussion of summer peak should be removed to the chapter Seasonality in Carbon.
>> This is removed.

Lines 321-326: the correlation coefficients have been presented in the table 2, do not repeat the values in the text. Concise these lines e.g. better correlations of TC, EC and OC were found between the protocols that use same POC correction method (DRI-TOR and IMPROVE_A TOR) than between Sunset-TOT, which use transmittance for POC correction and IMPROVE_A TOR (Table 2). Especially correlation of EC between Sunset-TOT and IMPROVE_A TOR was poor. Note, that Sunset-TOT and IMPROVE_A TOR had slightly different sampling time.
>> We accept the suggestion and this has been addressed.

Figure 6: CAPMoN time series have been already presented in Fig 4. Remove this figure and plot correlation plot between IMPROVE and CAPMoN 24h-measurements (, if exactly the same days are sampled) instead of monthly mean. Color-code the marks based on season/time/or something else.
>> Figure 4 was create to explain the gas adsorption artifact.  The CAPMoN time series were also included in Figure 6 as a direct visual comparison with IMPROVE and CABM measurements.  The reason why using monthly means throughout the entire paper has been addressed in the previous comment.

Lines 327-331 and Table 2: Clarify what kind of regression (linear, orthogonal) has been used. Prefer orthogonal. Clarify also, are the regression calculated from monthly mean values? Remove slopes, which are already presented in Table2. Explain why Regression 1 was used. Is it correct to force through the zero?
>> The information on the type of regression fit has been included.   Fitted parameters are included in a few places just to provide a quick reference to the readers.  Even though all fitted parameters are included in Table 2, having to look up values during reading can take some time.  The choice of linear regression fit is totally subject to the reader what method the reader may prefer.  Fitting the data through the zero is physically reasonable in many cases when we know an offset should not be present and the slope gives the best estimate of the relationship between the two sets of measurements.  In some situations, a non-zero intercept may also make sense as it may be physically be explained by over or under correction, or having a systematic bias.  That's why here we provide both sets of linear regression fit results so that readers can obtain the information they needed depending on what the reader may prefer to look for.

Lines 332-336: remove this paragraph to the new POC section.
>> We do not think a separate paragraph for POC is suitable.  As the reviewer suggested, POC is part of OC and we prefer to include POC discussion with other carbonaceous measurements.  In addition, POC is a charring correction under the IMPROVE or IMPROVE_A methods.  The ECT9 POC however is not a charring correction.  So we prefer not to directly compare the POC concentration from different protocol but just to point out their differences from the analysis.

**CABM vs. IMPROVE measurements**
Line 337: Change the subtitle e.g. Monthly comparison or something else
>> The subtitle has been removed as this section has been combined with the previous section.

As CABM measurements does not subtract the gaseous artefact, the writers may consider to plot figures 6 and 7 with uncorrected data.
>> The purpose of this analysis is to understand the difference in measurements among the various networks despite the unique differences in their sampling and analysis, which artifact correction is considered one of them.  By plotting the CABM measurements with the uncorrected CAPMoN measurements will only provide the relationship between the two data set. But it does not provide the information how the CABM measurements are compared with other measurements.

Figure 6: remove 6a-c to Supplement and delete 6d. Modify Fig 6a-c so that common x-axis is used to save space. Refer also to Figure 7 that should be presented against (x-axis) to CABM network that has the different sampling time compared to other networks.
>> Although it seems that the CAPMoN results are being shown twice (in Figure 4 and 6), the presence of the CAPMoN data are for different objectives.  In Figure 4 the data is shown for illustrating the magnitude of the gas adsorption artifact, whereas in Figure 6, we include the CAPMoN data for the completeness because that will give the reader a direct visual comparison of all the data from different

network. We have considered the suggestion to modify the x-axis of these time series graphs to save space but the results were not ideal. We have adjusted the size of the graph and try our best to make the graph clear.

Line 340-341: after "comparable" should be present correlation coefficient. The percentage shows the similarity of the concentrations. Also, if it is said that concentrations are higher, the writers should said where to compare "higher than". Modify this sentence.
>> The word "comparable" is removed and the sentence is modified.

Lines 342-346: Again, I do not understand why both regressions are presented. Why fits are forced through the origin? I recommend to use only regression with intercept unless there is a clear reason for forcing through the zero. Again if comparative is used, there have to be the other party.
>> The type of linear regression fit to use is really subjected to the reader preference. The authors believe the regression fit results forcing through zero is a good start of the analysis assuming there is no systematic bias or offset among the various data sets. In a few cases, we also extend our analysis to discuss results when not forcing the fit through zero. Tables 2 and 4 summarize all the linear regression fit results by forcing through zero and allowing an intercept.

Lines 351-352: CABM network did not see any short-term variation as it has week-long sampling time. Anyway the Fig. S3 shows monthly mean values that is even longer time than week. Modify the sentence.
>> The sentence is modified. Short-term variations are replaced by seasonal variations.

Line 353-359: POC discussion should be remove to its own section/paragraph. In line 356, Table 3 has not been presented yet. Why not use table 2? Remove the regression discussion with forced intercept
>> Table 3 summarizes the correlation coefficients among different variables (e.g., OC, EC, TC, etc.) and results have been used in various locations although we may not have explicitly mentioned in the manuscript. There are more discussion of the results summarized in Table 3 in the "normalized time series" section. However, since we are removing the normalized data analysis section, we moved Table 3 to the supplementary information.

**Comparison of the Normalized Time Series**
I do not understand this chapter. Why the data should be normalized to Jan 2008 data? This can be removed or explained better the meaning of this chapter.
>> We have removed the normalized analysis section and recombine some of the analysis to the ordinary inter-comparison section. Because of this, Figure 8 and 9 are also removed from the manuscript.

**Seasonality in Carbon**
Although this chapter if very interesting, it is not part of the objective. To stay with the objective, it would be interesting to see how the gaseous artefact correction varies between the season/temperature for DRI-TOR, Sunset-TOT and IMPROVE_A TOR. The writers can use the Sigmoid function if wanted but concentrate on the contribution of gaseous artefact. Also comparison of TC, OC and EC during different season between different networks is interesting. Is there any differences between different networks based on the season?
>> The authors think that the observed seasonality in carbon is an important observation that is result from the inter-comparison analysis. Therefore, the authors prefer to keep this section but has shorten it slightly. The authors have done separate analysis and it was observed that the POC from IMPROVE_A

TOR and DRI-TOR are always proportional to the OC because POC defined in this protocol is a charred OC correction.  Therefore, the seasonality observation for the IMPROVE_A TOR POC does not mean much as this, to certain extent, resemble the relationship seen in IMPROVE_A TOR OC.  The ECT9 POC, however, is different because our analysis and past research have shown that ECT9 POC represents separate category of OC compounds and therefore the seasonality relationship of the ECT9 POC actually provide additional insight that do not provide by OC and EC observations.

In the beginning of the Results and Discussion, the writers can present general overview of the results (yearly concentrations of TC, OC and EC). In addition, wind roses and footprints, if wanted, can be presented shortly here and put the plots in the Supplement.
>> The authors have removed the majority content regarding the wind rose and transport model results to the supporting materials.

**Conclusions:**
Based on the modification and comments, modify this chapter. In addition
Line 443-444: the filter face velocity does not affect for the field blank concentration. Now the readers may get wrong message. Modify. Add also information of the other field blanks e.g. field blanks accounted xx-xx% (xx-xx ugC/cm2) of the measured OC.
>> The sentence is modified to avoid confusion.

Lines 445-446: I am not sure if this statement was proved in this manuscript, although true. Modify this sentence e.g. Start with the information that CABM network did not correct gaseous artefact and its OC has xx% higher concentration than with two other networks that had the correction done.
>> This is addressed.

Lines 446-448: I do not understand this statement or its purpose. Too long story and too much details (like values of r). Describe the contribution of POC of TC /OC.
>> This is addressed.

Lines 451-452: SRM 8785 samples demonstrate the consistency of the different protocols not long-term carbon measurements. Correct.
>> This is addressed.

Line 457: "North American harmonized carbonaceous concentration map" is new for me and may be to other readers. Clarify shortly
>> The authors apologize for the confusion.  We have modified the sentence to better represent our true meaning.

**Supplement:**
Line 69-70. Actually internal signal is used to correct slight variation during each analysis. TC, OC and EC concentrations are calculated based on the calibration value calculated from external calibration.
>> The sentence is modified to improve clarity.

Correct. Line 79: There were no IMPROVE protocol. It was named to DRI-TOR and Sunset-TOT. Correct
Table S1: Use the protocols names you have chosen to use (Sunset-TOT and DRI-TOR). First column IMPROVE_A TOR, second column both Sunset-TOT and DRI-TOR and third column ECT9

>> The name "IMPROVE" and "IMPROVE_A" are the original name of the two protocols.  In this manuscript we have The paragraph has been modified to improve clarity.  Table S1 has also been updated to avoid confusion.

FigureS1: protocol name IMPROVE has been used, although not mentioned in the manuscript. Either rename that or add to the manuscript that the temperature steps used in analyzing particulate carbon in CAPMoN network are called IMPROVE although different optical correction used.
>> Figure S1 captions and figure content have been modified to be consistent with the rest of the manuscript.

Table S1 and Figure S1: Replace IMPROVE_A to IMPROVE_A TOR
>> This is addressed.

Figure S3: Add information that results are monthly averages.

>> This is addressed.

---

## Author Comment (AC2) · 18 Jul 2019

Review of "Inter-comparison of the elemental and organic carbon mass measurements from three North American national long-term monitoring networks" by Chan et al. This paper summarizes collocated organic and elemental concentrations from three different types of analysis and sampling protocol. The results are useful in furthering our understanding of thermal optical analyses and resulting biases from sampling artifacts as well as temperature protocols. The paper is fairly well organized and written but could benefit from some clarification. I recommend publication after the authors address comments below.
>> The authors appreciate the useful comments and suggestions from the referee, and we address all the comments accordingly.

Line 1: The title, as well as some description in the text is somewhat misleading because it implies that large geographic scale comparisons are made when in fact the comparisons only exist at one site. Perhaps changing or including something regarding different analytical protocols would help clarify this point.
>> The authors have modified the title to reflect the fact that the inter-comparison was done in one "co-located" site. Corresponding sections, including abstract and introduction have also been modified to reflect this information.

Line 23: Please state years.
>> This coverage information (2005-2015) has been included.

Line 23-26: Again, similar to the title, point out that collocated samplers only exist at one site, so really what is being compared here are the impacts from different sampling and analytical protocols, not a large scale geographic comparison.
>> This is addressed.

Line 29: More on this later, but I don't understand the value of the normalized comparison. The agreement depends on what you have normalized each time series to. Over what time periods where these comparisons made?
>> In the original study, each time series is normalized to its corresponding concentration measured on Jan 2008. This converts all concentration time series to a percentage change with respect to the measured concentration on Jan 2008. The comparisons for normalized concentration were made from 2008 to 2015.

After some consideration and discussions among the co-authors, we agree to remove the normalized comparison section and focus on the comparison on the absolute data.

Line 35: Is there any evidence for linkages to forest fire emissions and increased vehicular emissions? Did the authors include analysis of these emissions or is this conjecture?
>> Due to the length limit, it is not possible to include all the analyses in this manuscript. A separate analysis that involved the investigation of the 10 years BC emission trends at a number of CABM sites, including a boreal forest site, has suggested that the elevated BC emissions during summer at Egbert could be contributed by the forest fire emissions. The analysis is still currently on going and the full results is expected to be given in a separate manuscript.

Line 36-37: This may be true depending on artifact corrections and how they are applied across a network.

>> The abstract has been modified and this does not apply anymore as the original sentence was removed.

Line 38: Again, extrapolating data and comparisons from one site to "regional to continental-scale-harmonized maps" hasn't been shown here and may not necessarily be true, especially given different sampling times and sources.
>> The main idea of mentioning a "regional to continental-scale concentration map" is trying to express the effort of evaluating the consistency and compatibility from difference datasets by different networks when using atmospheric OC and EC measurements to constrain their emission changes at regional and continental scales. The "regional to continental-scale concentration map" is not proper and concise expression. It has been removed from the revised version.

The authors have included additional text in the introduction of the revised manuscript to explain the rational accordingly.

Line 23-41: I think it would be helpful if the abstract more closely reflected the comparison work rather than sources which was a rather minor part of the work and mostly based on previously published work (e.g., secondary aerosol formation in summer, smoke in summer, higher OC and EC in summer, etc.).
>> The abstract has been modified.

Line 52: OC can also absorb solar radiation.
>> The word "primarily" is now added.

Line 52,53,56,57: I would suggest using either BC or EC and keeping the same nomen-clature throughout the paper, unless the authors are actually referring to different measurements, then clarify and define.
>> BC and EC share lots of similarity in describing the physical appearance of the aerosol. In some cases, the word BC and EC can be inter-changed but not in all. We do not agree that all the term "BC" in the manuscript can be replaced by "EC" without changing the original meaning. The usage of BC and EC has been addressed in "introduction" of the revised version, and the authors have clearly defined the definition according to Petzold et al (2013).

Line 58: Include "impacts of" changing emissions since OC and EC measurements don't directly determine emissions.
>> This is addressed.

Line 61: The first sentence is unclear. Wouldn't long term measurements just depend on making the same measurement over time and doesn't really depend on a universal definition?
>> In fact, it is challenging to make ambient BC measurements. The word "long-term" is used in this manuscript because the main focus of this work is on long-term measurements. The corresponding sentence has been modified to avoid confusion.

Line 67: The sentence starting with "BC is a generic term" would be a better starting sentence for this paragraph and the authors could remove the current first sentence.
>> This is addressed.

Line 71: Replace "being" with "is"
>> This is addressed.

Line 72: Include "as" after "EC is referred to"
>> This is addressed.

Line 75: Can the authors clarify what they mean by "EC and BC resembled each other"?
>> We meant the trends in long-term time series of EC and BC concentration resembled each other. The sentence is now revised.

Line 80: I am not sure what the authors mean by "direct measurement of carbon mass as part of gravimetric mass"?
>> We meant the carbon mass measured by TOA or TEA is part of the particulate matter mass. This sentence is revised.

Line 97: I'm not sure what is meant by "resulting EC method"?
>> This sentence is revised.

Line 111: Can the authors provide a reference for the OC overestimation?
>> A reference is added.

Line 120: The acronyms for the various networks should be spelled out at first usage.
>> The acronyms of the three networks were first spelled out in the abstract. They are spelled out again in the revised version when it first appear after the abstract.

Line 122: Again, this is somewhat misleading. Add that these collocated measurements occurred at one site.
>> This is addressed.

Line 123: I might have missed this later, but what are the solutions for improving the compatibility?
>> The authors thank the referee for pointing this out and the word "solution" should not be used in here. Instead, the authors have replaced this by "suggestions". Based on the current work, two suggestions are: (1) ensure maintaining the same SOP for sampling and analytical procedure for any lab to ensure internal consistency, (2) to establish or include the use of a reference material or calibration materials (as suggested by World Meteorological Organization scientific advisory group) during the inter-comparison study. These information has now be included in the revised manuscript.

Line 124: I am not sure the results from one site have been demonstrated to create a regional and continental scale harmonized carbon concentration data set.
>> The authors realized we may not be expressing ourselves clearly and led to misunderstanding. The abstract has now been modified to remove those sentences. The corresponding content in the introduction has also been modified to clearly express our meaning when we meant to create a combined data set.

Line 138: What is meant by "regional-scale monitors"? Do the authors mean that many samplers operate across the United States?
>> We refer this to regional-scale monitoring stations. This is corrected in the revised version.

Line 139: replace "understanding long-range transport" with "understanding long-term trends".
>> This is addressed.

Line 139-140: I suggest replacing the Malm 1989 reference with the Malm 1994 reference (Malm, W. C., J. F. Sisler, D. Huffman, R. A. Eldred, and T. A. Cahill (1994), Spatial and seasonal trends in particle concentration and optical extinction in the United States, J. Geophys. Res., 99(D1), 1347-1370)
>> This reference is added.

Line 143-144: The IMPROVE samplers typically sample midnight to midnight, was the sampler at Egbert running on a different schedule?
>> Yes, the IMPROVE samplers at Egbert was run on a different schedule and this has been confirmed by DRI.

Line 148: Are the filters shipped cold?
>> Yes, they are shipped in coolers with ice pack.

Line 155: Spell out CAPMoN.
>> This is addressed.

Line 159: Do the measurements include carbon at all of these sites as well?
>> Historically there have been a number of sites that carry carbon analysis. However, they have been slowly shut down and Egbert is the only site with the longest collection history. This information is included in the revised version.

Line 176: Also see Malm et al. (2001) for a discussion of sampling biases on OC and EC concentrations (Malm, W. C., B. A. Schichtel, and M. L. Pitchford (2011), Uncertainties in PM2.5 gravimetric and speciation measurements and what we can learn from them, J. Air & Waste Manage. Assoc., 61, 1131-1149, doi:10.1080/10473289.2011.603998.)
>> The reference Malm et al., 2011 is now included.

Line 179: Spell ECCC- Also, please choose notation, either CABM or ECCC. Both are used interchangeably throughout the paper and it is confusing.
>> ECCC is removed from the subtitle.

Line 184: replace "costal" with "coastal"
>> This is addressed.

Line 207: replace "measurements is" with "measurements are"
>> This is addressed.

Line 211: Include "an" between "uses" and "impactor"
>> This is addressed.

Line 212: Replace "Impactor" with "Impactors"
>> This is addressed.

Line 218-222: See the Malm et al., 2011 paper mentioned earlier.
>> The reference Malm et al., 2011 is now included.

Line 228: Can the authors provide some references for the multiple studies?
>> References are now included.

Line 232: Change "introduce" to "introduces"

>> This is addressed.

Line 236: I think you can remove "SRM 8785 & 1649a" from the section header.
>> This is addressed.

Line 246: No correlations are given in Figure 1. Also include figure parts in the text and include OC.
>> Correlations are now included in the figure as well as the text.

Line 248-9: Need figure parts for Figure 2 in the text too (e.g., Figure 2(a)-(d) shows TC, EC, OC, and EC/TC, respectively)
>> This is addressed.

Line 249-250: It is unclear what the authors mean by "Irrespective of data disparity"?
>> This is now removed.

Line 270: I am not sure this is clear: Do the authors just resample the high resolution data for different averaging times? When they say different data sets do they mean the same measurement just with different averaging times? Wouldn't you expect these to compare well? Or do they compare EC to the PSAP measurement? The figures have units of Mm-1, so it suggests that they either converted EC to absorption coefficient (if so, what absorption efficiency was used?). Please clarify, including figure caption 3 when "comparison of different sets of measurements" from (c) because it is misleading.
>> In this section, we use the 1 min resolution PSAP data (measuring aerosol absorption, assumed dominantly by BC) as a common data set. We then average this data set to the once every third day resolution to simulate IMPROVE and CAPMoN data. We also average the 1 min PSAP data to weekly integrated values to simulate the CABM data. The reason we do not directly compare IMPROVE or CAPMoN data with CABM is because these measurements were not in same sampling frequency and therefore when converting these data to monthly averages, there is no way to know if any difference in monthly means was caused by the natural data variations in the original measurements or it was due to the difference in sampling frequency (it could be caused by both factors). In addition, this analysis was done by Yang et al (2011) and therefore it is not repeated in this manuscript. To ensure no obvious bias caused by the difference in sampling frequency, two different monthly means of PSAP (by every three days vs. by weekly integrated) are directly compared. We have clarify the paragraph in the revised version.

Line 255: Which EC/TC value was further verified? Also, replace "sample" with "samples"
>> In Figure 2d (the original Figure 1d), the three blue bars represent the EC/TC ratio reported by the certificate and also determined from the inter-comparison from the TEA and TOA methods. The green bar represent the EC/TC value calculated from an independent method based on carbon isotope. Here we mean to verify the EC/TC values determined from the TEA/TOA method by carbon isotope method.

Line 283: Over what time period?
>> This is addressed.

Line 288: Can the authors comment on the offset (nonzero intercept) and what it implies in terms of sampling artifacts or biases?
>> A linear regression fit forcing through the origin was applied to Figure 5. The authors believe a fit through zero makes more sense because any non-zero intercept would imply that the

artifact correction obtained from the backup filter was either too much or not enough compared to the actual artifact. The fact that the intercepts were insignificant suggests this is a reasonable assumption and the artifact correction was reasonable.

Line 298: Yes, the POC correction directly influences EC concentrations. Can the authors comment on this vapor adsorption issue with respect to the PSAP weekly comparisons?
>> PSAP in an in-situ instrument that continuously measures the changes in the amount of light transmitted through a quartz filter when particles are deposited onto the filter inside the PSAP. Even though filter media is involved in PSAP measurements, vapor adsorption is not expected to be an issue for PSAP measurements because there is no heating involved, so the adsorbed materials do not char and contribute to absorption.

Line 303: Add a period and start "An optical correction" as a new sentence.
>> This is addressed.

Line 317: Include "monthly mean" before DRI-TOR CAPMon measurements and "comparable to the concentrations derived from the IMPROVE_A: : :."
>> This is addressed.

Line 320: What are considered "good correlations"?
>> We consider measurements with correlations above 0.8 to be a good correlation.

Line 351-352: Can the authors describe Figures 7a-c before 7d to keep them in order?
>> This is addressed.

Line 355: At what level of significance?
>> Here the significant correlation is a relative comparison based on the correlation coefficient. We have corrected the wording in the sentence to avoid confusion.

Line 358: I am not convinced the normalized analysis is necessary and adds to the paper. The comparisons between samplers would change depending on what the data are normalized to (choose a different month or an annual mean for example). The comparisons already discussed are more useful because they show the true biases. The diurnal wind cycles on the timelines could be added to the earlier timelines if the authors want to include that analysis.
>> We agree with the referee and we have removed the normalized analysis section and combine some of the information into the section where inter-comparison of the absolute measurements. Because of this, Figure 8 and 9 are now removed from the manuscript.

Line 393: I think elevated carbon concentrations in summer are better shown in Figure 6 given the averaging times.
>> We have modified the sentence to reference this.

Line 413: When are the concentration in the N and NW higher?
>> For OC, elevated concentration could occur during SOA formation when air mass is originated from the N and NW. For EC, elevated concentration could potentially be related to forest fire emissions although more research is needed to verify this.

Line 414: Do the authors mean residential instead of residual?
>> Thank you, and this is addressed.

Line 431: How appropriate is the comparison with ECT9 POC since this is a nonlinear relationship?
>> The authors do not totally understand this comment.  However, the corresponding text has been revised to avoid confusion.

Line 447: Also include longer sampling time.
>> This is addressed.

Line 452: What are typical measurement uncertainties? Are these greater?
>> Typical uncertainties could be about 15% for individual OC and EC measurements.  The monthly averages should be higher than 20%.

Line 452: Note that others have performed similar comparisons across networks (CSN and IMPROVE) for continental scale integration. Biases for both OC and EC between networks were less than 10% (similar sampling and analytical procedures). Hand, J. L., B. A. Schichtel, M. Pitchford, W. C. Malm, and N. H. Frank (2012a), Seasonal composi-tion of remote and urban fine particulate matter in the United States, J. Geophys. Res., 117, D05209, doi:10.1029/2011JD017122. Hand, J. L., B. A. Schichtel, W. C. Malm, and N. H. Frank (2013), Spatial and temporal trends in PM2.5 organic and elemental carbon across the United States, Advances in Meteor., 2013, Article ID 367674.
>> The reference has been included accordingly.

Line 504: This link did not work, it needs to be updated:
http://vista.cira.colostate.edu/Improve/improve-data/
>> The authors cannot locate the above link. We believe there was a mistake for not copying the proper link. The authors have checked the link (http://vista.cira.colostate.edu/improve/Data/QA_QC/Advisory.htm ) in the acknowledgement section and ensure it is working.

Line 699: Table 1. Can the authors clarify: Is "IMPROVE" under CAPMoN consistent with lines 170-171 that lists IMPROVE-TOT for 2005-2007 and IMPROVE_TOR protocol? It is challenging to keep these different protocols straight and so careful attention to how they are referred to in the paper and the tables would help.
>> The authors understand the concern from the referee.  We have modified the names of the protocols throughout the paper to ensure they are consistent.

Line 702: Table 2: Similar comment, here it is referred to as "Sunset-TOT". The number of significant digits included in this table seem unnecessary.
>> The protocol name has been verified to be consistent with other parts of the manuscript.  We keep the additional significant digits to ensure no round off error when those information will be used by the readers.

Line 710, 713: I don't think these tables are necessary, see earlier comment.
>> This table has now been removed from the main paper and be included in the supplementary information.

Line 718: Figure 1: Again, please be consistent with ECCC and CABM
>> CABM (Canadian Aerosol Basement Measurement) is our network name whereas ECCC (Environment and Climate Change Canada) is our institution name. We believe ECCC is more appropriate in Figure 1 (now become Figure 2) caption.

Line 725: Figure 2, Same comment as previous figure. What is ICP? Please relate x-axis labels to the caption description.
>> This is addressed.

Line 732: Figure 3: See earlier comment- this comparisons is unclear.
>> The corresponding paragraphs have been modified to provide additional information to explain these figures.

Line 740: Figure 4: It would help to see the comparisons in (b) and (c) if the scales were reduced. Again, note the data description in the figures do not match the discussions or tables (e.g., "Sunset-TOT")
>> We have modified the names of the protocols throughout the paper to ensure they are consistent with the description in the Figure caption.

Line 757: Figure 6: Please include location in this figure caption so it is clear that the three different networks are collocated at one site.
>> This is addressed.

Line 766: Figure 7: Which "IMPROVE" are the comparisons made against? Please be clear in the caption to match the axis labels.
>> This is addressed.

Figures 8 and 9: Are unnecessary and do not lend to a better understanding of the comparisons.
>> These figures are now removed.